# Measuring CLEVRness: Black-box Testing of Visual Reasoning Models

**Spyridon Mouselinos**
University of Warsaw
Warsaw, Poland
s.mouselinos@uw.edu.pl

**Henryk Michalewski**
University of Warsaw, Google
Oxford, U.K.
henrykm@google.com

**Mateusz Malinowski**
DeepMind
London, U.K.
mateuszm@deepmind.com

## Abstract

How can we measure the reasoning capabilities of intelligence systems? Visual question answering provides a convenient framework for testing the model's abilities by interrogating the model through questions about the scene. However, despite scores of various visual QA datasets and architectures, which sometimes yield even a super-human performance, the question of whether those architectures can actually reason remains open to debate. To answer this, we extend the visual question answering framework and propose the following behavioral test in the form of a two-player game. We consider black-box neural models of CLEVR. These models are trained on a diagnostic dataset benchmarking reasoning. Next, we train an adversarial player that re-configures the scene to fool the CLEVR model. We show that CLEVR models, which otherwise could perform at a "human level", can easily be fooled by our agent. Our results put in doubt whether data-driven approaches can do reasoning without exploiting the numerous biases that are often present in those datasets. Finally, we also propose a controlled experiment measuring the efficiency of such models to learn and perform reasoning.

## 1 Introduction

Are our artificial intelligence systems capable of reasoning? Or like *Clever Hans*, they use various cues only tangentially related to the task and rely on rote memorization with poor generalization? (Pfungst, 1911; Johnson et al., 2017a) This work revisits such a question and proposes an interactive framework with the communication channel between two players. The first player, which reasoning capabilities we are about to test, performs visual reasoning tasks, we call it *Visual-QA Player*. The second player, which we call the *Adversarial Player*, is manipulating the scene so that it *fools* the first player even though those changes still lead to correct reasoning steps among humans. Both players interact with each other only through questions, answers and the visual scene as shown in Figure 1. If the *Adversarial Player* manipulating the scene causes the *Visual-QA Player* to change its answer even though the new scene is still valid for the same question and answer, it is then the reasoning failure. It is similar to the following situation. Imagine a box is placed between two spheres. If you ask a question, *is there a box between two spheres?*, the answer should be positive. Now, if we move the box anywhere so it does not cross any of the spheres, and ask the same question, the response should remain unchanged. In other words, we postulate that reasoning outputs of agents need to be invariant under scene configurations that are consistent with the questions-answer pairs. Moreover, in the spirit of generic adversarial attacks, we seek configurations that also pose little if any reasoning challenges for humans.

We propose an automatic and agnostic pipeline to benchmark the reasoning capabilities of various models, only assuming they communicate by answering questions about the scene. Due to the recent stream of research in vision-and-language (Zhang et al., 2021; Jiang et al., 2020; Guo et al.,

2019b; Su et al., 2020; Wang et al., 2020; Kamath et al., 2021; Tan & Bansal, 2019; Chen et al., 2020), we believe there will be an increasing number of vision models that operate through language. Moreover, we also consider the visual question answering framework set-up as a two-player system as an excellent benchmarking pipeline. We perform all tests by scene manipulations and observing how a tested model behaves under such changes. The pipeline does not require any knowledge of the internals of the tested model. It also does not manipulate the sensory information of such a model, e.g., pixels in the images, and all the manipulations are physically meaningful. Even though our current pipeline uses synthetic scenes as only those can easily be automatically manipulated, our results have also real-world ramifications. If models are susceptible to semantically *meaningless* changes[1] in scene configurations, in a synthetic setting, there are valid concerns that real-world robots could also be prone to manipulation of objects in a room. Finally, our work also questions the possibility of training and benchmarking networks in a purely data-driven and offline, static manner.

**Contributions.** The main contributions of our work could be summarized in three points.

***First***, we propose a strong *black-box* adversarial test, which makes no assumptions about the under-lying mechanics of a tested model, formulated as a game between two players. Our test does not require any direct access to the tested model, even through its sensory information. In particular, it does not require gradients, output probabilities, or any access to the perceived image. Our work also deviates from bounded perturbations and instead focuses on global scene manipulations that are still consistent with the task constraints, and can change the behavior of a tested model.
***Second***, we reformulate visual reasoning by integrating visual question answering with zero-sum two-player game frameworks. Under our novel formulation, a visual and adversary agents compete against each other through content manipulation. We believe that this is an initial step towards more sophisticated frameworks that integrate computer vision with multi-agent systems.
***Third***, we explore the limits of the data-driven approaches in synthetic visual scenarios, and demonstrate that current CLEVR models are lacking the efficiency to learn robust reasoning steps.

## 2 RELATED WORK

Our work touches upon various research directions, which we briefly describe here.
**Visual QA.** Introduced as a visual counter-part of the Turing Test (Malinowski & Fritz, 2014; Geman et al., 2015), it became a computer vision task that requires a holistic visual understanding. Many other works have extended the task to larger datasets or videos or adversarial train-test splits (Antol et al., 2015; Agrawal et al., 2018; Tapaswi et al., 2016; Yu et al., 2019). More recently, we also observe the generalization of the task to become a part of the vision-plus-language suite of benchmarks (Lu et al., 2019; Chen et al., 2019; Wang et al., 2021). Johnson et al. (2017a) have introduced a synthetic variant of the Visual QA problem that is more focused on reasoning and the control of the experimentation. Although it was shown that traditional methods (Yang et al., 2016) are not enough to solve that dataset, newer methods can pass the human baseline on this task. These methods are trained to operate on pixels and text directly (Santoro et al., 2017; Perez et al., 2018; Hudson & Manning, 2018; Malinowski et al., 2018; Kamath et al., 2021) or they induce programs as an intermediate representation (Johnson et al., 2017b; Mascharka et al., 2018; Yi et al., 2018). Hudson & Manning (2019) have shown that Visual QA models lack some reasoning robustness but their approach is mostly linguistic and still static. Some other works also observe issues with static evaluation and proposed more dynamic benchmarks, e.g., with a human-in-the-loop (Khashabi et al., 2021; Nie et al., 2020; Li et al., 2021; Sheng et al., 2021). We extend the line of research on Visual QA by reformulating it as an interactive two-agents system, and show that CLEVR under such a new setting is still an unsolved problem. Our work also differs from previous works in that it focuses on visual reasoning, is interactive and fully automatic.
**Adversarial perturbations.** Szegedy et al. (2014); Goodfellow et al. (2015); Kurakin et al. (2016); Carlini & Wagner (2016) have introduced early methods that can 'fool' networks by performing tiny, visually imperceptible, perturbations of the input image. Moosavi-Dezfooli et al. (2017) have also shown universal and untargeted adversarial perturbations that are more transferable than previous approaches. Most methods perform *white-box* attacks, where an adversarial model has access to the target model's parameters or gradients. In *black-box* attacks such access is removed and, e.g., only output probabilities are available (Guo et al., 2019a). Other *black-box* attacks involve the use

---

[1]Changes that are consistent with the task constraints.

of surrogates to estimate gradients of the real target (Papernot et al., 2017; Cheng et al., 2019) or gradient-free methods (Alzantot et al., 2018). More related to our work, Cheng et al. (2018) describe a *black-box* system, which only assumes access to outputs of the network. However, all the perturbations described above are continuous-based where individual pixels are changed independently. This leads to improbable images and assumes access to the sensory inputs of the neural network. In contrary, *semantic* perturbations (Joshi et al., 2019; Zeng et al., 2019), operate on semantically meaningful chunks of the input, leading to more plausible and physically interpretable perturbations. However, they still assume direct access to either images or ideally differentiable renderers. We can interpret our work within the adversarial perturbations framework, where the adversarial model can change the original scene by its manipulations, with zero access to the target model, including its sensory information and not limited to tiny changes but instead to semantically and globally *meaningless* modifications.

**Reinforcement learning.** The game between the *Visual-QA Player* and *Adversarial Player* is a zero-sum two-player game. Conneau et al. (2017) use a conceptually similar two-player game to find a better word alignment between two languages without parallel corpus. Inspired by such a problem formulation, we consider our two-player game as a tool to achieve better reasoning models. We optimize the *Adversarial Player* with Advantage Actor-Critic (A2C) algorithm (Sutton & Barto, 2018; Degris et al., 2012).

**Probing and measuring intelligence.** Turing Test (Turing, 2009) is considered to be among the earliest works on measuring the intelligence of artificial systems. Crucially to us, it is also formulated as a two-player game with an interrogation protocol. Legg (2008) systematizes and relates the notion of intelligence from different fields with the main focus on 'universality'. Johnson et al. (2017a) have transferred the notion of intelligence onto the visual ground and emphasized reasoning rather than universality. It seems there is no widely accepted dataset or definition that encompasses our intuitions about intelligence. However, there are a few recent directions showing the lack thereof, mainly highlighting bias amplification or absence of mathematical capabilities (Hendricks et al., 2018; Bhardwaj et al., 2021; Piekos et al., 2021).

## 3 PRELIMINARIES

In this section, we explain briefly how the CLEVR dataset (Johnson et al., 2017a) is constructed and introduce our notation and definitions.

**CLEVR** is a synthetic visual question answering dataset introduced by Johnson et al. (2017a), which consists of about 700k training and 150k validation image-question-answer triplets. Images are artificially constructed and rendered from scene graphs – a special structure containing information about object attributes such as position or color. Such a scene graph is also used to synthesize the ground-truth question-answer pairs by expanding templates according to the depth-first-search ordering. Ambiguous scenes are rejected. Each image represents an isometric view of the scene containing from two to ten objects. There are three classes of objects, *spheres*, *cubes* and *cylinders*. Each object can also be either large or small and has one color out of four (brown, purple, cyan, yellow). It can also be either metallic or rubber-made. Every object has $x$ and $y$ coordinates that are confined within the $(-3, +3)$ range. We use the same generation process to render modified scenes.

**CLEVR models.** Various models have been introduced to work with the CLEVR dataset, some even 'solving' the dataset by achieving near perfect performance. Despite the strong offline performance, we test if those models' performance perpetuates in the more interactive setting where configurations of the scene could be changed. Whenever possible, we use pre-trained CLEVR models. Otherwise, we train the remaining models from scratch by making sure we achieve results similar to published accuracy numbers on the validation set. We summarize all the models in Table 1. We show the accuracy on the CLEVR dataset (*Accuracy*), indicate if an architecture is trained from scratch (*Re-trained*), briefly describe how multi-modal fusion and reasoning is conducted (*Reasoning Mechanism*), and indicate any extra privileged information required during the training process (*Extra*). For instance, some models require extra access to functional programs used during the dataset generation, use scene graphs as a supervisory signal (states), or always operate on scene graphs (input-states). Otherwise, the models were trained only from image-question-answer triples.

***Mini-games.*** We formulate our problem as a *Game* between two players, *Visual-QA Player* and *Adversarial Player*. The *Visual-QA Player* takes as input question-image pairs and provide answers to such questions. Some models use states (scene-graphs) that replace images or require programs (Johnson et al., 2017a). The whole game consists of all CLEVR data points. For our

purpose, we extend the notion of the *Game* into *Mini-games*. The rules of *Mini-games* are identical to the whole *Game*. The only difference is that each *Mini-game* operates on a subset of the CLEVR dataset. We define the size of a *Mini-game* by the number of datapoints that are attached to that *Mini-game*. We sample data points for each *Mini-game* randomly and mutually exclusively. *Mini-games* have analogies in the adversarial perturbations literature. *Mini-games* of size one resemble per-image adversarial perturbations (Goodfellow et al., 2015; Moosavi-Dezfooli et al., 2016) whereas a *Mini-game* that has all data pointsis similar to universal adversarial perturbations (Moosavi-Dezfooli et al., 2017). In this work, we investigate various *Mini-game* sizes but due to the sheer scale we were unable to use the whole game as the *Mini-game*. Larger *Mini-games* make the optimization process more difficult as the domain where the *Adversarial Player* needs to operate increases. The training is also much more time-consuming and a sequential process. Instead, we can train multiple players on different *Mini-games* independently and thus massively. We leave the arduous training of the universal *Adversarial Player* on the whole *Game* as a possible future direction.

Table 1: CLEVR models that we use as *Visual-QA Players*.

| Model Name | Accuracy | Re-trained | Reasoning Mechanism | Extra Data |
|---|---|---|---|---|
| SAN (Yang et al., 2016) | 72.1 | | Attention | |
| FiLM (Perez et al., 2018) | 96.2 | ✓ | Feature Conditioning | |
| RN (Santoro et al., 2017) | 93.2 | ✓ | Relational | |
| IEP (Johnson et al., 2017b) | 96.9 | | Neural Program Induction | Programs |
| TbD (Mascharka et al., 2018) | 99.1 | | Neural Program Induction | Programs |
| Mdetr (Kamath et al., 2021) | 99.7 | | Multimodal Transformer Querying | States |
| State-Input Transformer (ours) | 96.8 | ✓ | Cross Attention | Input-States |

## 4 ENVIRONMENT

We need to ensure that *Adversarial Players* create valid scenes that are *consistent* and *in-distribution*. Both properties are guaranteed by our environment enforcers.

**Consistency.** Since scene manipulation may change the answer for a given question, we need to ensure this does not happen. That is, the new scene is still *consistent* with the question-answer pair. The question-relevance enforcer achieves that by running functional programs associated with each question (Johnson et al., 2017a) on the modified scene-graph. Hence, it gets the new ground-truth answer. The enforcer rejects the new scene if that new answer differs from the previous one. In this way, it guarantees the newly generated scenes give the same answers as the original scenes on the same question. Thus, we can generate equivalent scenes containing the same objects that have identical answers for the same questions. Using that enforcer, we can test if the *Visual-QA Player*'s answers are invariant under such an equivalent class of scenes.

**In-distribution.** Even with the question-relevance enforcer, the *Adversarial Player* may still produce undesired outputs. For instance, it can stretch the whole scene thus violating the scene boundaries from the original CLEVR dataset, making, e.g., everything to look very small (Section A.9 in the appendix). Although it is still an interesting form of adversarial scene manipulation, we focus rather on the *in-distribution* scene manipulations that respect the original boundaries. To enforce that property, we use a scene-constraint enforcer that checks the boundaries of the scene. Without that enforcer, the *Adversarial Player* would quickly resort to stretching the whole scenes, achieving a form of adversarial attack that uses distribution shifts rather than content manipulation. It does so, e.g., by moving the camera away until objects are barely visible. We give a few such examples in the appendix (Section A.9).

## 5 *Adversarial Player*

Meaningful scene manipulations require not only generic scene understanding, but also the ability to distinguish which objects to displace and how. Hence, the player is a composition of a multi-modal module, which creates input representation, and a decision maker, which decides how to control the scene. Figure 1 illustrates the *Adversarial Player* and the game between both players.

**Multi-modal module.** We have experimented with the same multi-modal modules as those in Table 1, but found out we have a better performance and the convergence rate if the *Adversarial Player*

operates on the scene-graphs (states) instead of pixels. For that, we use *state-input* variant of Relation Networks (Santoro et al., 2017). The model receives as the input $10 * 6$ object tokens, and question tokens. Every object token represents one-out-of-ten possible objects in the scene by its attributes such as position, color, shape, material and size. If the scene has fewer than ten objects, we use $\emptyset$ token to indicate that, which also acts as a padding. We also have special tokens that separate questions from the objects which we add as a latent embedding, e.g., $\text{emb}(\text{material}) + \text{emb}(\text{object})$. Such an input encoding is similar to our *State-Input Transformer* and described in Section A.3. The embedded vectors are given to the Relation Network (RN). Finally, we train that network on the CLEVR visual question answering task, where we achieve $97.6\%$ on the validation set, and use the representation just after the last relational layer for the decision maker.

**Decision maker.** Inspired by the work on reinforcement learning (Mnih et al., 2016; Wu et al., 2017; Lillicrap et al., 2016; Wang et al., 2016; Schulman et al., 2015), we use an actor-critic module that acts on scenes. The actor is a general-purpose fully connected layer with ten object-specific heads. Each head is randomly assigned to a unique object in the scene for its manipulation. Every head produces a displacement in $x$ and $y$ coordinates of the corresponding object. Although we have initially experimented with the continuous output space, we have found out the following simple strategy is more effective. First, we discretize all the $x$ and $y$ coordinates into $N$ bins each. Now, each head produces two $N$-dimensional vectors that are next projected into a probabilistic space via softmax. Next, we sample displacements in $x$ and $y$ axis independently from both softmax distributions. Note that, even though we do not model the joint distribution explicitly due to computational reasons, both samples condition on the common head and thus are only *conditionally* independent of each other. We discretize the scene where each axis has values in $[-3, 3]$ onto $N = 7$ bins per axis. Our critic is a simple three layer feed-forward network (with relu as activations) that predicts a reward score between $-1$ and $+1$ via tanh activation ($1.2 * \tanh$ for better numerical properties).

**The game of scene manipulations.** Due to our formulation of *Adversarial Player* and the environment, we can benchmark various reasoning models purely in the *black-box* setting via a series of questions about the scene. *Adversarial Player* manipulates the scene so that it is still consistent with the question-answer pair. The manipulations are applied to scene-graphs, and the resulting scene-graph is evaluated by the environment enforcers described in Section 4. Invalid scenes are thus discarded. In this way, we ensure the *in-distribution* and *consistency* in the scene generation. Original image-question pairs are fed to a *Visual-QA Player* that produces corresponding answers. We refer to that answers as *old answers*. After the scene manipulation, new images paired with the same questions are also given to the *Visual-QA Player* that produces *new answers*. We construct rewards based on *old answers*, *new answers* and *ground-truth answers*.

If the *Adversarial Player* forces the *Visual-QA Player* to change the answer, i.e., an *old answer* is different than a *new answer*, it gets *Consistency Drop Reward* (*cr*). If an *old answer* is the *ground-truth* answer, it gets instead *Accuracy Drop Reward* (*dr*). Both rewards differentiate between simply confusing the model and causing a drop in its performance. If *Adversarial Player* produces an

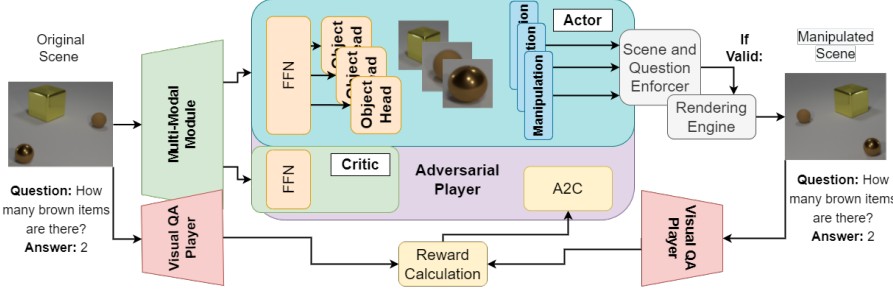

Figure 1: Our game between two players: *Adversarial Player* and *Visual-QA Player*. *Adversarial Player* uses a *multi-modal module* to extract features conditioned on the visual and textual inputs. After transforming such features with a feed-forward architecture, it samples an action using object-specific heads. Each action corresponds to manipulating the corresponding object in the scene. In the case of missing objects, we use an $\emptyset$ token. After alternating the original scene graph, we use various environment enforcers to ensure validity of the constructed scene. A valid scene graph is rendered and introduced to the *Visual-QA Player* together with the original image. Finally, we collect responses of the *Visual-QA Player* and calculate suitable rewards based on them, and we repeat the whole cycle during the training phase.

invalid scene it gets *Invalid Scene Reward* (*isr*). This reward encourages producing scenes that pass the environment enforcers tests. Finally, if *Adversarial Player* does not manage to fool the model, it gets *Fail Reward* (*fr*). We use the following values: $dr= 1, cr= 0.1, fr = -0.1, isr = -0.8$.

**Training algorithm.** To train *Adversarial Player* we use the A2C algorithm with the episode length set to one as we do not need to model long-range consequences of the decision-making mechanism. Batches contain images, question, answers, programs and scene-graphs. We train the *Adversarial Player* for each *Mini-game* independently using the same architecture. We experiment with the following *Mini-game* sizes $10, 100, 1000$. All *Mini-games* are constructed randomly. Under our discretization scheme, the action space is $N^k$ where $N$ is the number of bins and $k$ is the number of objects in the scene. In practice, it is up to $49^{10}$.

## 6 EXPERIMENTS

We consider *Adversarial Player* with a multi-modal module pre-trained either on states (*state-input*) or directly from pixels (*pixel-input*). We train *Adversarial Player* for each *Mini-game* independently. We use different *Mini-game* sizes in our experiments: 10, 100 and 1000. To obtain statistically significant results, we run each *Mini-game* thirty times with randomly initialized *Adversarial Players*, each with different seed per trial. We report the results that are averaged over all those runs. We compute two metrics for each *Mini-game*. *Consistency* refers to the fraction of times *Adversarial Player* has changed the *Visual-QA Player*'s answer, regardless if that answer was correct. *Drop* refers to the fraction of times *Adversarial Player* has changed the *correct Visual-QA Player*'s answer. We aggregate the results using two statistics. *Average Accuracy* averages accuracies over all *Mini-games*. *Maximal Accuracy* refers to the worst (best) case of a single *Mini-game* from the *Visual-QA Player*'s (*Adversarial Player*'s) perspective. We also computed $p$-values of T-Test, with the null hypothesis that a manipulation is unsuccessful. In almost all cases we reject the null hypothesis with small $p$-values. The $p$-values are available in the appendix (Section A.11).

**Quantitative results.** We show our results in Table 2 for the *state-input* and *pixel-input Adversarial Players*, and for each *Visual-QA Player* presented in Table 1. All the *Visual-QA Players* in the comparison are trained from pixels, apart form our custom *State-Input Transformer* architecture (Section A.3 in the appendix). When comparing both *Adversarial Players*, we can observe that the *state-input Adversarial Player* is significantly more successful than the *pixel-input* one. We hypothesize that this is the effect of having more structured and less ambiguous input information. It could also be the case that multi-modal modules trained from pixels ignore features in the same way and hence smaller discrepancy between all models operating on pixels. Moreover, as the size of *Mini-games* increases, it is becoming more difficult for the *Adversarial Player* to successfully manipulate the model on all examples from the *Mini-game*. This is an expected behavior resembling standard adversarial perturbations (Goodfellow et al., 2015) and their universal variants (Moosavi-Dezfooli et al., 2017). The highest performing CLEVR models, TbD and MDetr, are the most robust under the manipulation. However, they are still susceptible to the scene manipulations in some *Mini-games* as the *Maximal Consistency* metric indicates, especially if our *Adversarial Player* uses states as the input. Section A.8 (appendix) also shows richer performance statistics. Our detailed investigation shows that the models above are particularly sensitive to manipulations of scenes associated with *counting* or *existence* questions. Such question are often more complex, hence, increasing the likelihood of the reasoning failure at some stage (Section A.6 in the appendix).

**Qualitative results.** Figure 3 shows qualitative results of the scene manipulations for the three highest performing and distinctive models from Table 1; e.g., simple feed-forward or program induction models. For the reader's convenience, we provide in Section A.2 (appendix) examples of CLEVR object attributes. We can observe that the scene manipulations are surprisingly minimal and are semantically *meaningless*, i.e., they should not 'fool' human players.

**Reasoning steps and visual cues.** To better understand the source of errors (reasoning steps vs visual cues), we have conducted the following experiments. In the first experiment, shown in Section B.4 (appendix), we visualize the attention mechanisms of *Visual-QA Players* before and after the scene manipulations. We observe changes in the attention map of all models but MDetr. The results suggest, most models's perception is impacted by the manipulations. However, all objects are still correctly identified by MDetr, thus suggesting some issues are also stemming from the reasoning steps. In the second experiment, we deployed Slot-attention object detector (Locatello, 2020), and show that the detector identifies the same properties before and after manipulations. This result (Section B.6; appendix) shows that *Adversarial Player* does not produce 'corner cases' of the

Table 2: Results of the games: *Model dataset performance* refers to models' accuracy on the CLEVR dataset. *Model Mini-game performance* refers to the model accuracy on the *Mini-game* examples where the largest (maximal) performance drop was detected, before manipulations took effect. The *Average Accuracy* column reports the performance of models averaged over all runs of the respective *Mini-game* sizes. *Maximal Accuracy* reports the worst model performance among the respective *Mini-game* size runs. We also report in brackets the relative performance drop, in percentages $\frac{X-Y}{X}\%$. Average Accuracy is compared against *Model dataset performance* ($X$ = Model dataset performance / $Y$ = State/Pixel-Input). Maximal Accuracy is compared against *Model Mini-game performance* ($X$ = Model *Mini-game* performance, $Y$ = State/Pixel-Input). Note that in the case of state-input and pixel-input *Adversarial Player* the worst *Mini-game* might not come from the same *Mini-game* instance.

| Model | *Mini-game* size | | Average Accuracy | | | Maximal Accuracy | |
|---|---|---|---|---|---|---|---|
| | | Model dataset performance | State-Input | Pixel-Input | Model *Mini-game* performance | State-Input | Pixel-Input |
| SAN | 10 | | 61.8 (-14.2%) | 63.8 (-11.4%) | 80.0 / 80.0 | 28.0 (-65.0%) | 43.0 (-46.2%) |
| | 100 | 72.1 | 66.1 (-8.3%) | 69.2 (-3.9%) | 74.0 / 75.0 | 47.6 (-35.6%) | 66.3 (-11.5%) |
| | 1000 | | 70.2 (-2.5%) | 71.0 (-1.4%) | 72.3 / 72.3 | 68.4 (-5.3%) | 70.8 (-1.9%) |
| FiLM | 10 | | 83.9 (-12.7%) | 93.6 (-2.6%) | 100.0 / 100.0 | 48.0 (-52.0%) | 86.0 (-14.0%) |
| | 100 | 96.2 | 89.1 (-7.3%) | 94.8 (-1.4%) | 98.0 / 100.0 | 75.6 (-22.8%) | 92.4 (-7.5%) |
| | 1000 | | 93.8 (-2.4%) | 95.8 (-0.4%) | 96.4 / 96.1 | 90.8 (-5.7%) | 94.7 (-1.4%) |
| RN | 10 | | 80.5 (-13.6%) | 86.4 (-7.2%) | 100.0 / 100.0 | 47.0 (-53.0%) | 74.0 (-26.0%) |
| | 100 | 93.2 | 85.5 (-8.2%) | 90.8 (-2.5%) | 94.0 / 95.0 | 63.2 (-32.6%) | 87.4 (-7.9%) |
| | 1000 | | 90.5 (-2.8%) | 91.6 (-1.7%) | 93.1 / 93.3 | 90.0 (-3.2%) | 91.0 (-2.4%) |
| IEP | 10 | | 84.3 (-13.0%) | 94.4 (-2.5%) | 100.0 / 100.0 | 48.0 (-52.0%) | 87.0 (-13.0%) |
| | 100 | 96.9 | 90.3 (-6.8%) | 95.6 (-1.3%) | 98.0 / 96.0 | 74.4 (-24.0%) | 93.5 (-2.6%) |
| | 1000 | | 94.1 (-2.8%) | 96.6 (-0.3%) | 97.1 / 97.3 | 93.9 (-3.2%) | 95.7 (-1.6%) |
| TbD | 10 | | 94.0 (-5.1%) | 99.1 (-0.0%) | 100.0 / 100.0 | 69.0 (-31.0%) | 99.0 (-1.0%) |
| | 100 | 99.1 | 96.6 (-2.5%) | 98.7 (-0.4%) | 100.0 / 100.0 | 91.1 (-8.8%) | 98.0 (-2.0%) |
| | 1000 | | 98.0 (-1.1%) | 99.0 (-0.1%) | 99.4 / 99.7 | 95.5 (-3.9%) | 98.8 (-0.9%) |
| Mdetr | 10 | | 93.7 (-6.0%) | 99.7 (-0.0%) | 100.0 / 100.0 | 60.0 (-40.0%) | 99.0 (-1.0%) |
| | 100 | 99.7 | 96.1 (-3.6%) | 98.6 (-1.1%) | 100.0 / 100.0 | 86.4 (-13.5%) | 97.8 (-2.2%) |
| | 1000 | | 98.5 (-1.2%) | 99.4 (-0.3%) | 99.5 / 100.0 | 94.5 (-5.0%) | 99.1 (-0.9%) |
| State Input Transf. | 10 | | 89.3 (-7.7%) | 96.0 (-0.8%) | 100.0 / 100.0 | 77.0 (-23.0%) | 91.0 (-9.0%) |
| | 100 | 96.8 | 94.7 (-2.1%) | 96.1 (-0.7%) | 97.0 / 99.0 | 92.6 (-4.5%) | 95.0 (-3.9%) |
| | 1000 | | 95.7 (-1.1%) | 96.5 (-0.3%) | 97.2 / 96.4 | 95.1 (-2.1%) | 95.8 (-0.5%) |

perceptual system. We also have similar conclusions based on our small-scale human experiment (Section B.5; appendix).

**Adversarial training.** We have adapted a widely used adversarial training schema (Goodfellow et al., 2018), and included manipulated scenes in the training protocol of a *Visual-QA Player*. We use *pixel-input* RN (Santoro et al., 2017) as the *Visual-QA Player*. Using this method, we have obtained only marginal improvements in robustness at the cost of a slight performance degradation on the original CLEVR dataset.

**Limitations of data-driven reasoning models.** Due to the data-driven nature of our *Visual-QA Players*, we pose the following question, How many examples of *Adversarial Player* manipulations would be enough for a visual reasoning model to train on, in order for it to be robust against any unseen ones? Intuitively, if the network has enough capacity, and was trained on all possible data points (or its manipulations), it could rely on a look-up strategy to solve the problem accurately. We propose the following experiment to address such a question. We created a series of datasets as follows. We treat a scene as a discrete $7 \times 7$ grid (manipulations are restricted only to this grid). Thus each dataset contains $49^2 = 2401$, $49^3 = 117649$, $49^4 = 5764801$ scene combinations, covering all possible scene manipulations with two, three, and four objects respectively. In the case of four objects, due to computational reasons, we keep a single object stationary (thus it also has $49^3$ scene manipulations). The set of questions associated with each image dataset requires the model to perform either one reasoning step (Onehop), two reasoning steps (Twohop) or are a mixture of both (Mixhop). We use RN, FiLM and TbD as *Visual-QA Players*. We train them on $X\%$ data and next evaluate on $(100-X)\%$ remaining, unseen data. We have conducted ten trials by forming $X\%$

training data randomly. We notice that the behavior of all tested models is similar so we report a joint average. Results are presented in Figure 2. As the number of objects and diversity of question increases, so does the amount of training examples needed in order to achieve robustness.

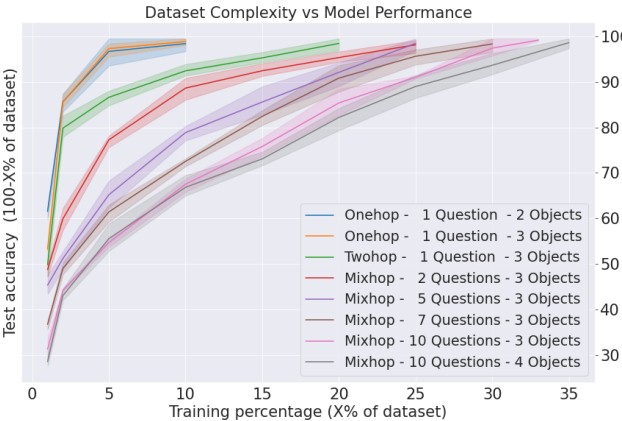

Figure 2: Data-driven limitation experiment: Unseen perturbation accuracy depends heavily on the number and complexity of introduced questions, followed by the number of objects in the dataset examples/scenes. A model could be considered robust when achieving close to 100% test accuracy.

## 7  CONCLUSIONS

We present a novel approach to isolate and benchmark the reasoning capabilities of visual models by formulating a game between two players. Both players are independent and can only communicate through a common interface, an environment and question-answer pairs. One player, named *Adversarial Player*, is trying to manipulate the environment so that the second player, named *Visual-QA Player*, can't answer the given question anymore. We show that popular models with solid results on CLEVR are susceptible to scene manipulations, and their performance may degrade significantly. Moreover, we also conducted a controlled study on the generalization capabilities of such models. We did so by creating synthetic and simplified datasets of all possible scene manipulations. We show that the existing models are susceptible to reasoning gaps and require extra data, proportional to their task complexity in order to generalize to all possible configurations. One possible direction to increase efficiency could be the use of stronger inductive biases. Another to change the training paradigm into a more interactive one. However, the environment is synthetic, and it could be that learning from real-world and multi-modal data is somewhat more efficient. Finally, we believe that melding two-player games with the visual question answering framework is both natural and beneficial for testing the reasoning capabilities and potentially can also be generalized to other settings.

### ACKNOWLEDGMENTS

All experiments were performed using the Entropy cluster funded by NVIDIA, Intel, the Polish National Science Center grant UMO-2017/26/E/ST6/00622 and ERC Starting Grant TOTAL. The work of Spyridon Mouselinos and Henryk Michalewski was supported by the Polish National Science Center grant UMO-2018/29/B/ST6/02959.

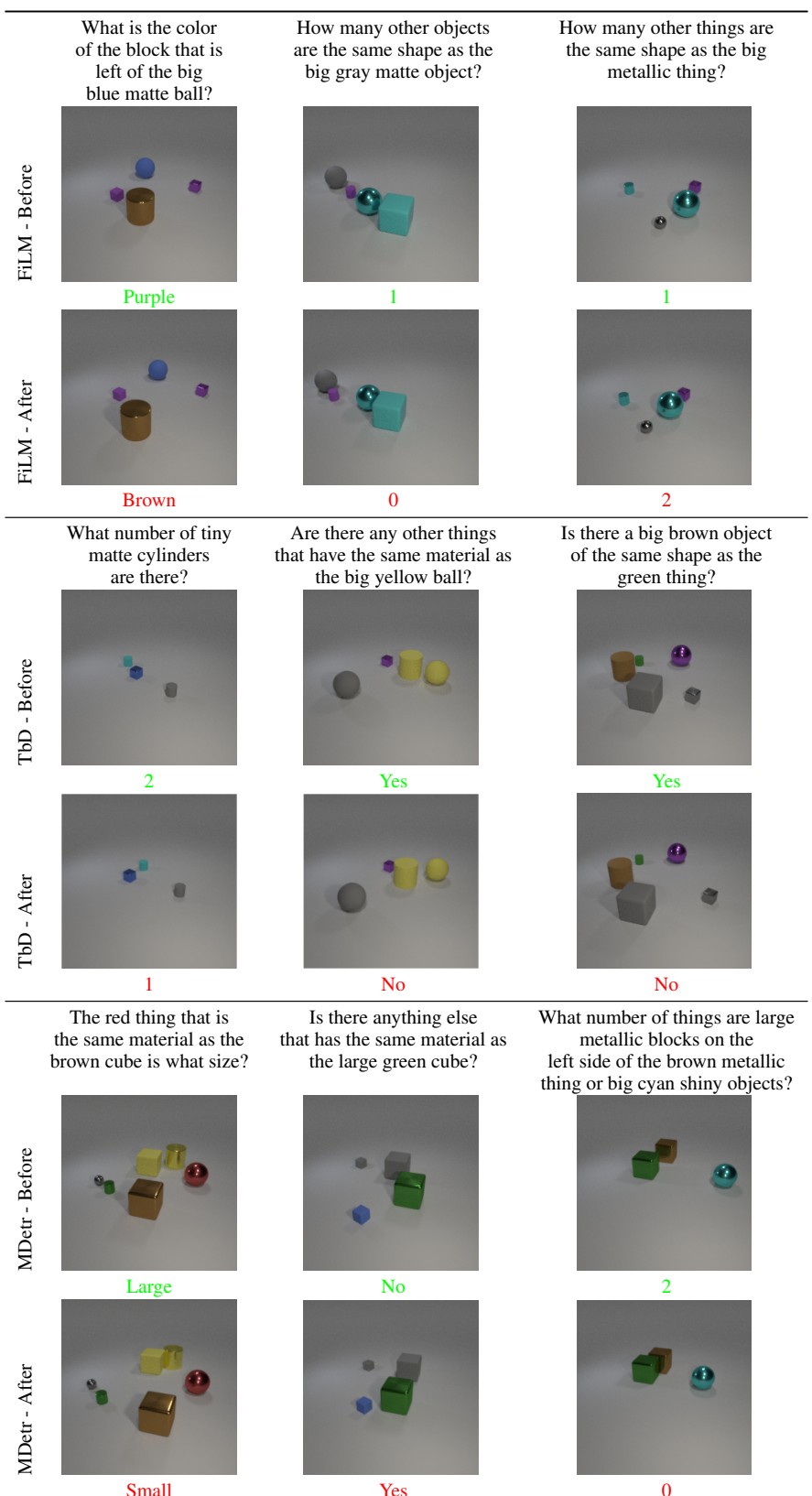

Figure 3: Manipulations of CLEVR models. We show results *before* and *after* scene manipulations.

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

# A   APPENDIX

## A.1   URLS OF CLEVR MODELS

Table 3 shows the URLs to models used in our investigations (also Table 1 in the main paper). We also report if we re-trained a model from scratch (type *Architecture*) or used already trained models (type *Model*). Please note that the latter type proves that our testing procedure is fully black-box.

| Model Name | Link | Type |
|---|---|---|
| SAN (Yang et al., 2016) | https://github.com/facebookresearch/clevr-iep | Model |
| IEP (Johnson et al., 2017b) | https://github.com/facebookresearch/clevr-iep | Model |
| FiLM (Perez et al., 2018) | https://github.com/ethanjperez/film | Architecture |
| RN (Santoro et al., 2017) | https://github.com/mesnico/RelationNetworks-CLEVR | Architecture |
| TbD (Mascharka et al., 2018) | https://github.com/davidmascharka/tbd-nets | Model |
| Mdetr (Kamath et al., 2021) | https://github.com/ashkamath/mdetr | Model |

Table 3: URLs to Models. *Architecture* denotes that we use the code but re-trained the model on CLEVR. *Model* refers to already trained models.

## A.2   OBJECT CATEGORIES

Figure 4 shows a made-up image that contains all available CLEVR object categories: shapes, sizes, and materials. Small objects are rendered at approximately $40\%$ of the size of their large counterparts. Metallic materials can be identified by the high albedo property and 'shininess' of the object. We provide it as the reference point for a reader to make it easier to compare our qualitative results.

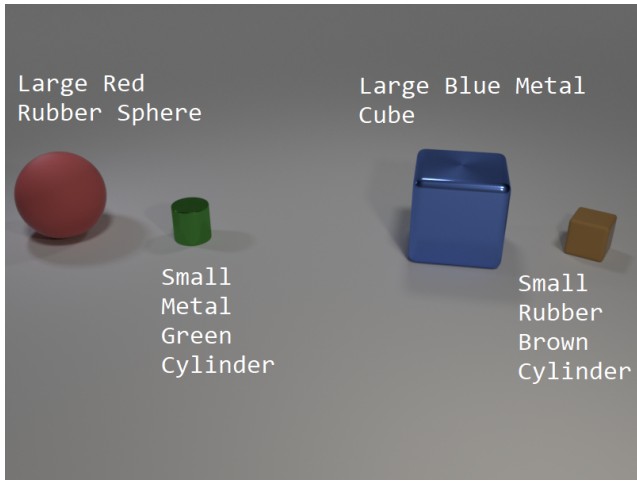

Figure 4: Object categories in the CLEVR dataset.

### A.3    STATE-INPUT TRANSFORMER

To test our two-player game against a *state-input Visual-QA Player*, we have designed a transformer-based architecture (Vaswani et al., 2017) that receives six types of input features – object sizes, object shapes, object materials, object colors, object positions, and question tokens – and uses a cross-modal attention mechanism. In that mechanism, queries from one modality attend to keys and values from the other modality. Each data point contains variable-length inputs, describing all objects in the scene. It also contains question tokens that compose the question itself. Hence, we do padding with a special token $\emptyset$ to the maximal input length. We set the maximal length for the objects tokens to be $10 * 6 = 60$ and $50$ for the question tokens. All in all, we have $110$ input tokens. Each token type is projected into a different embedding space and a learnable type embedding is added to it, separately for object and question tokens. For instance, $\mathrm{emb}(\mathrm{material}) + \mathrm{emb}(\mathrm{object})$. Furthermore, three special learnable embeddings are used as an additional input. We use them as queries to reduce the overall computational costs of the cross-attention mechanism. The same mechanism is also used in Perceiver (Jaegle et al., 2021). The input tokens (concatenated object and question tokens) form keys and values. In every transformer block, we apply cross-attention between all input embeddings and those three special tokens. This is repeated five times. As the final block, a feed-forward network (classifier) receives as inputs the three latent tokens and outputs answer probabilities. We show that architecture in Figure 5.

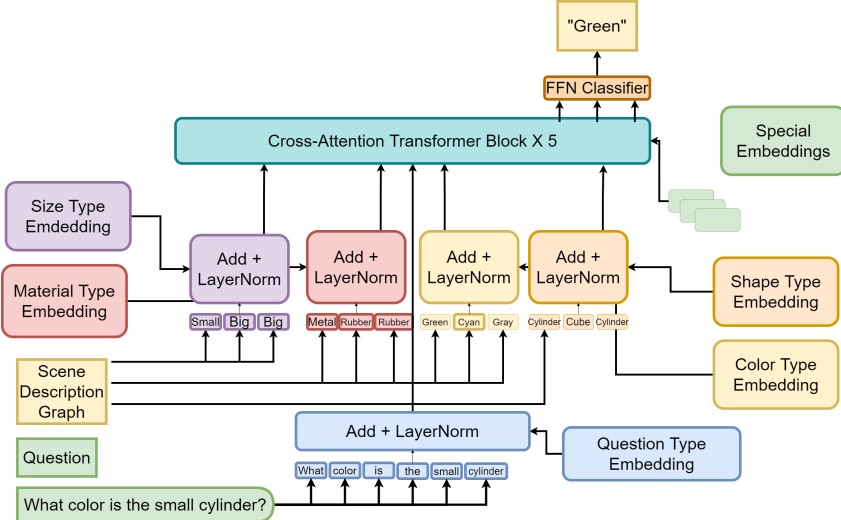

Figure 5: Our State-Input Transformer that operates on the graph scene (states).

### A.4    MANIPULATIONS OF *State-Input Visual-QA Player*

Figures 6 and 7 show manipulations of *Visual-QA Player* that is trained to take *states* instead of *pixels* as the input. Such a *state-input Visual-QA Player* receives a direct overview of the scene, bypassing any need for any image renderings. We use *State-Input Transformer* (Section A.3) as the *multi-modal* component of that player. Since such a model gets the perfect visual information as the input, it makes it more robust under scene manipulations. As a consequence, our *Adversarial Player* tends to manipulate the scene so that objects are placed closely together.

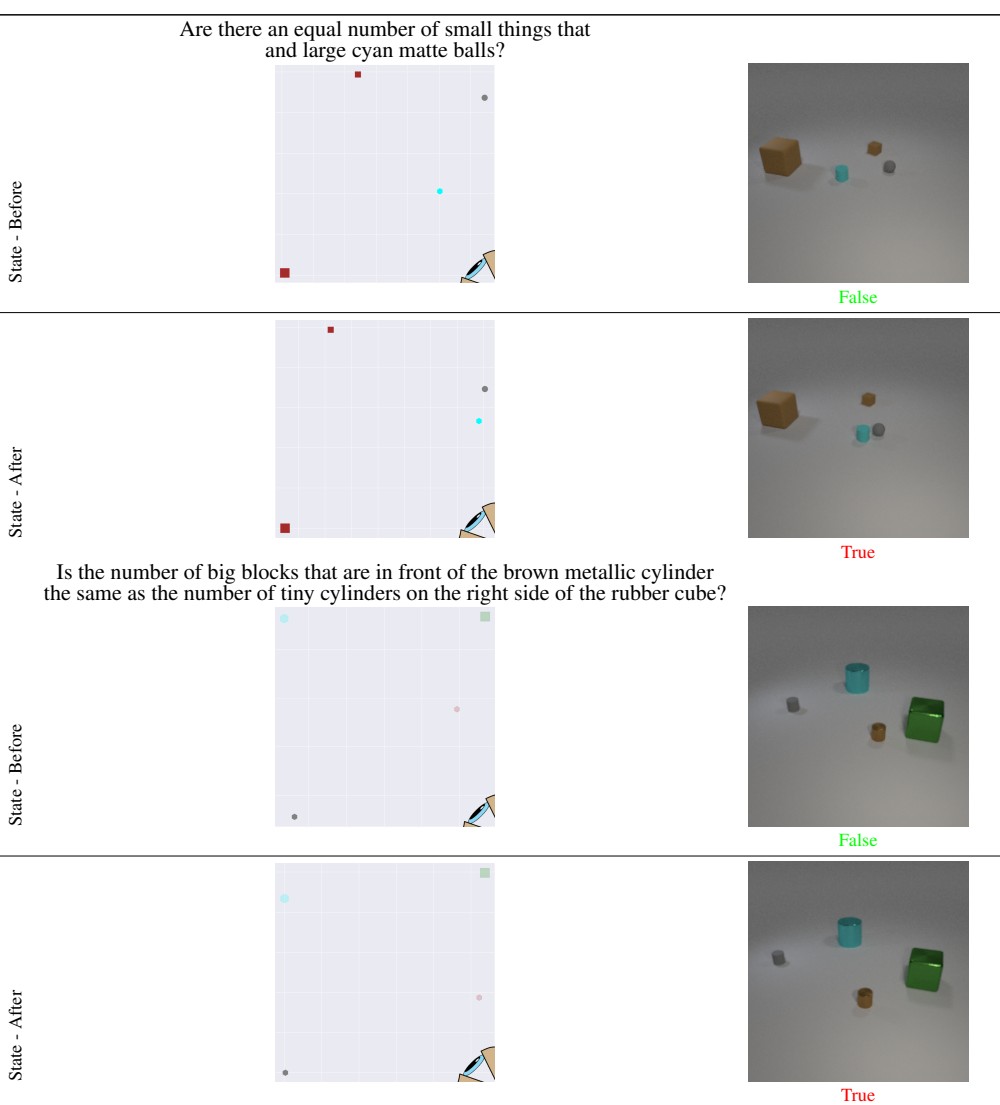

Figure 6: Qualitative results on *state-input Visual-QA Player*. For each example, a top-down view of the scene is presented. The eye at the bottom right of each view represents the rendering camera. It corresponds to the *Visual-QA Player*'s viewpoint. Each scene is also rendered and presented side-by-side for comparison.

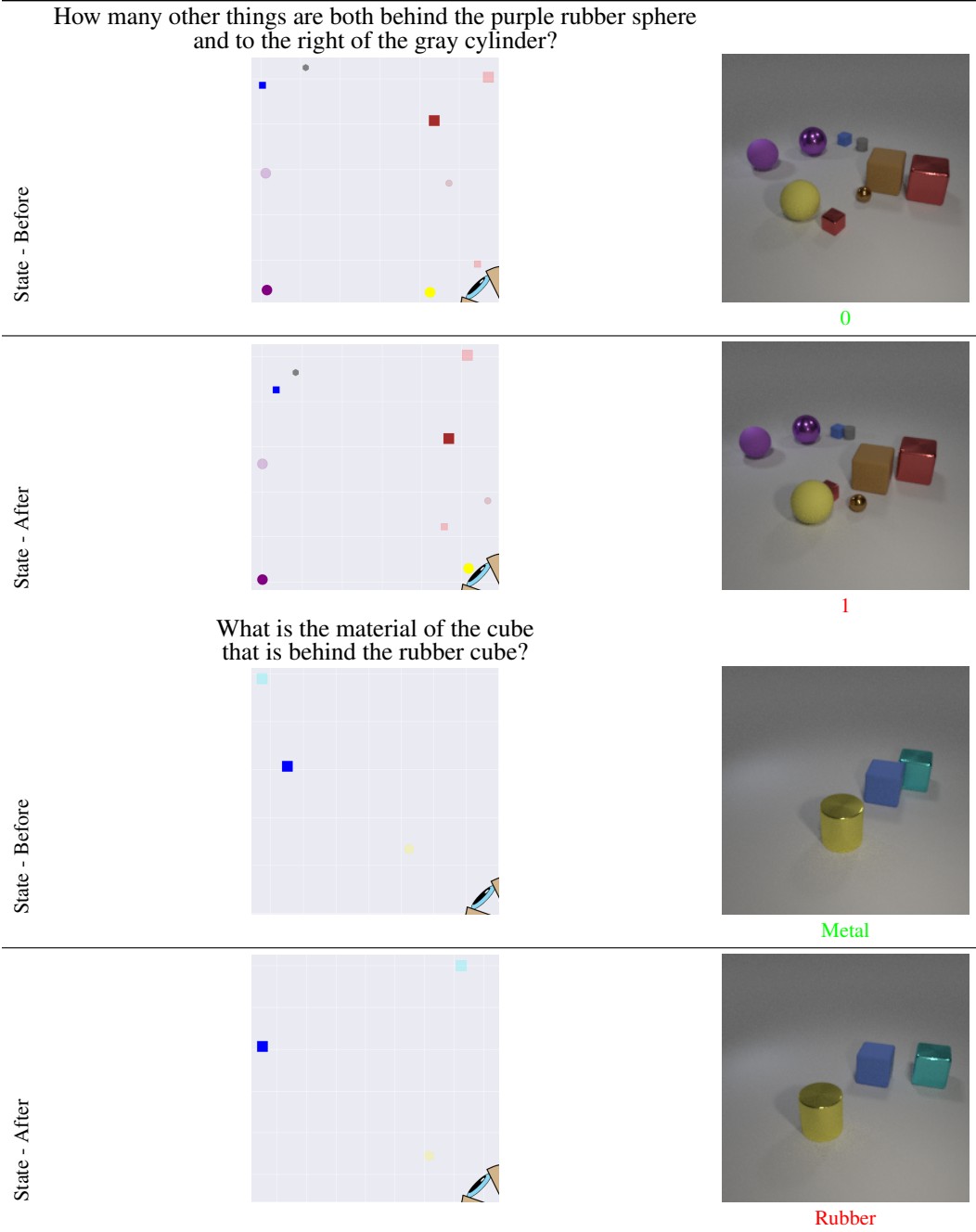

Figure 7: Qualitative results on *state-input Visual-QA Player*. For each example, a top-down view of the scene is presented. The eye at the bottom right of each view represents the rendering camera. It corresponds to the *Visual-QA Player*'s viewpoint. Each scene is also rendered and presented side-by-side for comparison.

## A.5 ALGORITHMS

We show pseudo-algorithms that we use to (Algorithm 1) calculate rewards, (Algorithm 2) train *Adversarial Player*, (Algorithm 3) and play a game.

---

**Algorithm 1** Calculate Rewards

---

1: $new\_answer$ : Answer produced by Visual Agent on perturbed image
2: $old\_answer$ : Answer produced by Visual Agent on original image
3: $gt\_answer$ : Ground Truth Answer
4: $dr$ : Drop Reward
5: $cr$ : Consistency Reward
6: $fr$ : Fail Reward
7: $reward \leftarrow 0$
8: **if** $new\_answer \neq old\_answer$ **then**
9:     **if** $old\_answer \neq gt\_answer$ **then**
10:         $reward = reward + cr$
11:     **else**
12:         $reward = reward + dr$
13:     **end if**
14: **else**
15:     $reward = reward + fr$
16: **end if**
17: **return** $reward$

---

---

**Algorithm 2** Training Pipeline

---

1: $M$ : Minigames
2: $Va$ : Visual Agent, $Fe$ : Feature Extractor
3: $A$ : Actor, $C$ : Critic
4: $Qre$ : Question Relevance Enforcer, $Sce$ : Scene Constraint Enforcer
5: **for** $batch\ in\ M$ **do**
6:     $rewards, state\_values, logprobs \leftarrow DataGame(M, Va, Fe, A, C, Qre, Sce)$
7:     $n \leftarrow |rewards|$
8:     $advantage \leftarrow rewards - state\_values$
9:     $ploss \leftarrow -logprobs \times advantage$
10:     $vloss \leftarrow (state\_values - stop\_grad(rewards))^2/n$
11:     $loss \leftarrow ploss + vloss$
12:     $backprop(loss, A, C)$
13: **end for**
14: **return**

---

---

**Algorithm 3** Game between players (Game)

---

1: $M$ : *Mini-game*
2: $\mathcal{P}_{\text{VQA}}$ : Visual Agent, $\mathcal{S}_k$ : Multi-modal Backbone
3: $A$ : Actor, $C$ : Critic
4: $Qre$ : Question Relevance Enforcer, $Sce$ : Scene Constraint Enforcer
5: $isr$ : Invalid Scene Reward
6: **for** $batch\ in\ M$ **do**
7:     $rewards \leftarrow \{\}$
8:     $state\_values \leftarrow \{\}$
9:     $logprobs \leftarrow \{\}$
10:     **for** $(image, scene, question, program, gt\_answer)\ in\ batch$ **do**
11:         $old\_answer \leftarrow \mathcal{P}_{\text{VQA}}(image, question)$
12:         $extracted\_features \leftarrow \mathcal{S}_k(scene, question)$
13:         $dx\_probs, dy\_probs \leftarrow A(extracted\_features)$
14:         $CatX \leftarrow Categorical(dx\_probs)$
15:         $CatY \leftarrow Categorical(dy\_probs)$
16:         $state\_value \leftarrow C(extracted\_features)$
17:         $dx \leftarrow$ sample $CatX$
18:         $dy \leftarrow$ sample $CatY$
19:         $new\_scene \leftarrow Perturbate(scene, dx, dy)$
20:         $check\_scene \leftarrow Sce(new\_scene)$
21:         $check\_question \leftarrow Qre(new\_scene, program, gt\_answer)$
22:         **if** $check\_scene \wedge check\_question = True$ **then**
23:             $new\_image \leftarrow Render(new\_scene)$
24:             $new\_answer \leftarrow \mathcal{P}_{\text{VQA}}(new\_image, question)$
25:             $rewards \leftarrow rewards \,||\, CalcRewards(new\_answer, old\_answer, gt\_answer)$
26:         **else**
27:             $rewards \leftarrow rewards \,||\, isr$
28:         **end if**
29:         $state\_values \leftarrow state\_values \,||\, state\_value$
30:         $logprob \leftarrow logprob \,||\, log(p(dx|CatX))) + log(p(dy|CatY)))$
31:     **end for**
32: **end for**
33: **return** $rewards, state\_values, logprobs$

---

## A.6 SENSITIVITY TO QUESTION TYPES

Figure 8 shows the susceptibility of CLEVR models to scene manipulations of *Adversarial Player*. We observe that especially *counting* and *existence* questions are the 'back-doors' for our scene manipulations. Questions of those two types typically involve multiple steps of reasoning. *Adversarial Player*'s scene manipulations make them more likely that reasoning will fail at some stage.

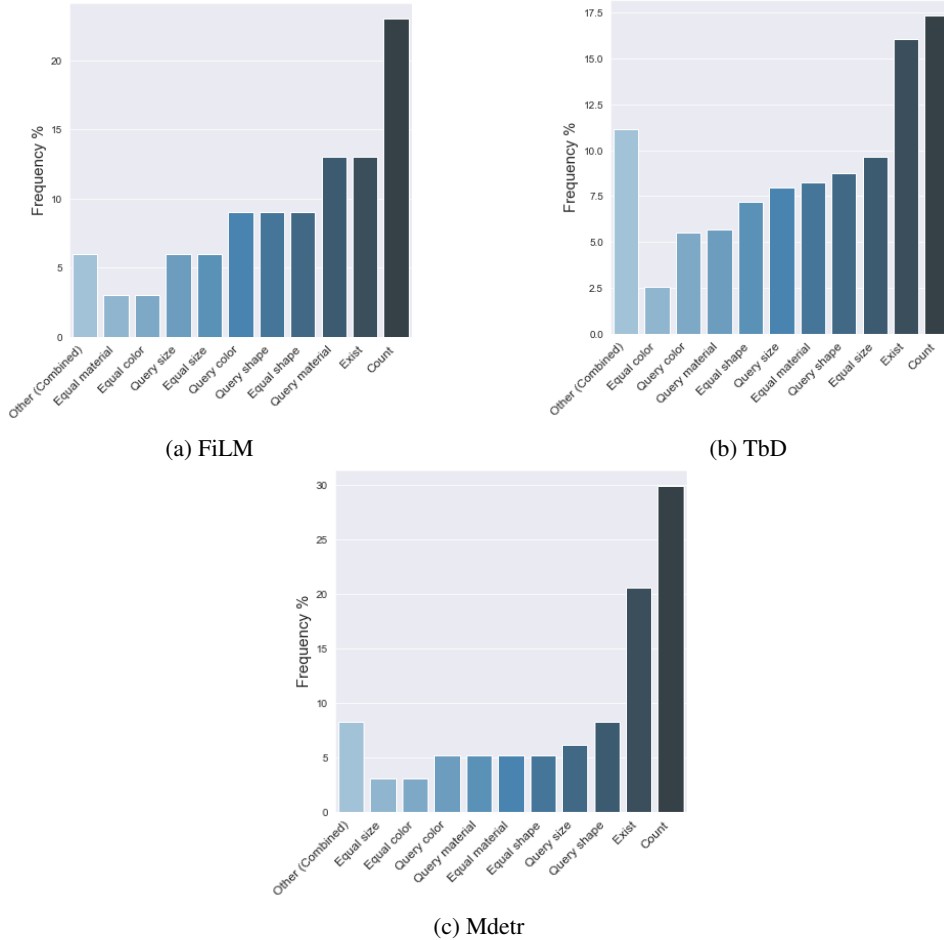

(a) FiLM                                      (b) TbD

(c) Mdetr

Figure 8: Histograms of the model's susceptibility to scene manipulations conditioned on a question type.

## A.7 CONVERGENCE

Figure 9 shows convergence plots of *Adversarial Player* during training against FiLM, TbD and MDetr. We use different initialization seeds and 30 trials. *Adversarial Player* is trained either from *states* or *pixels*. The former yields higher performance and sometimes better convergence. We use the *Drop* metric that measures the accuracy drop after the *Visual-QA Player* is manipulated. The higher *Drop*, the more successful manipulations are.

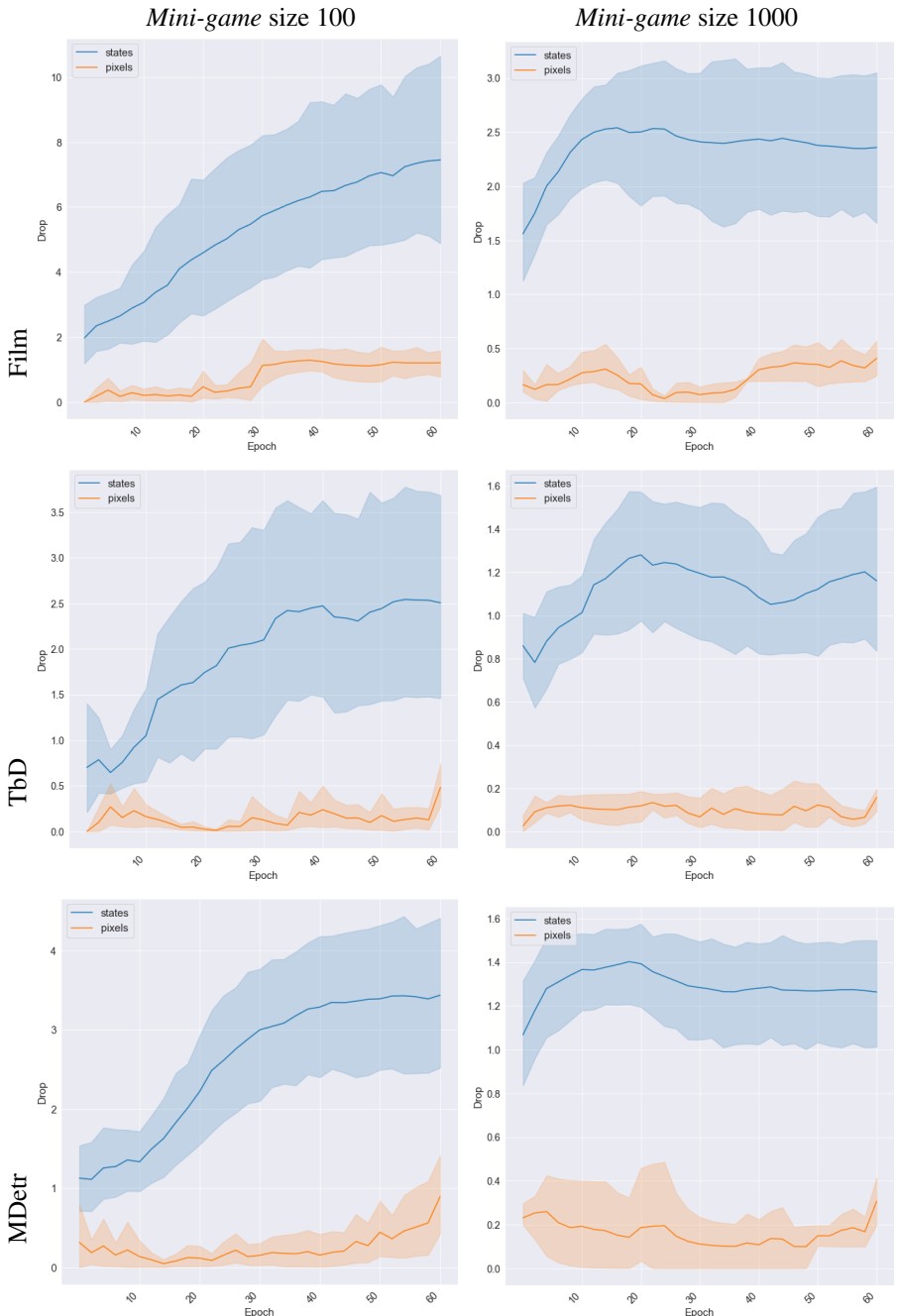

Figure 9: Performance of *Adversarial Player* for different training snapshots. We show the mean and variance over thirty trials. *Adversarial Player* is either trained from *states* (blue) or *pixels* (orange).

## A.8 PERFORMANCE DISTRIBUTION PLOTS

Figures 10 and 11 show the distribution of the *Adversarial Player* performance, where we show how often (y-axis) the given *Drop* score is achieved (x-axis). The *Drop* metric measures the accuracy drop after the *Visual-QA Player* is manipulated. The higher *Drop*, the more successful manipulations are. Larger *Mini-games* lead to more consistent (narrower distributions) but lower performance. This behavior suggests that scene diversity present in larger *Mini-games* leads to more stable scene manipulations. However, they do so at the cost of increasing the complexity of the optimization problem.

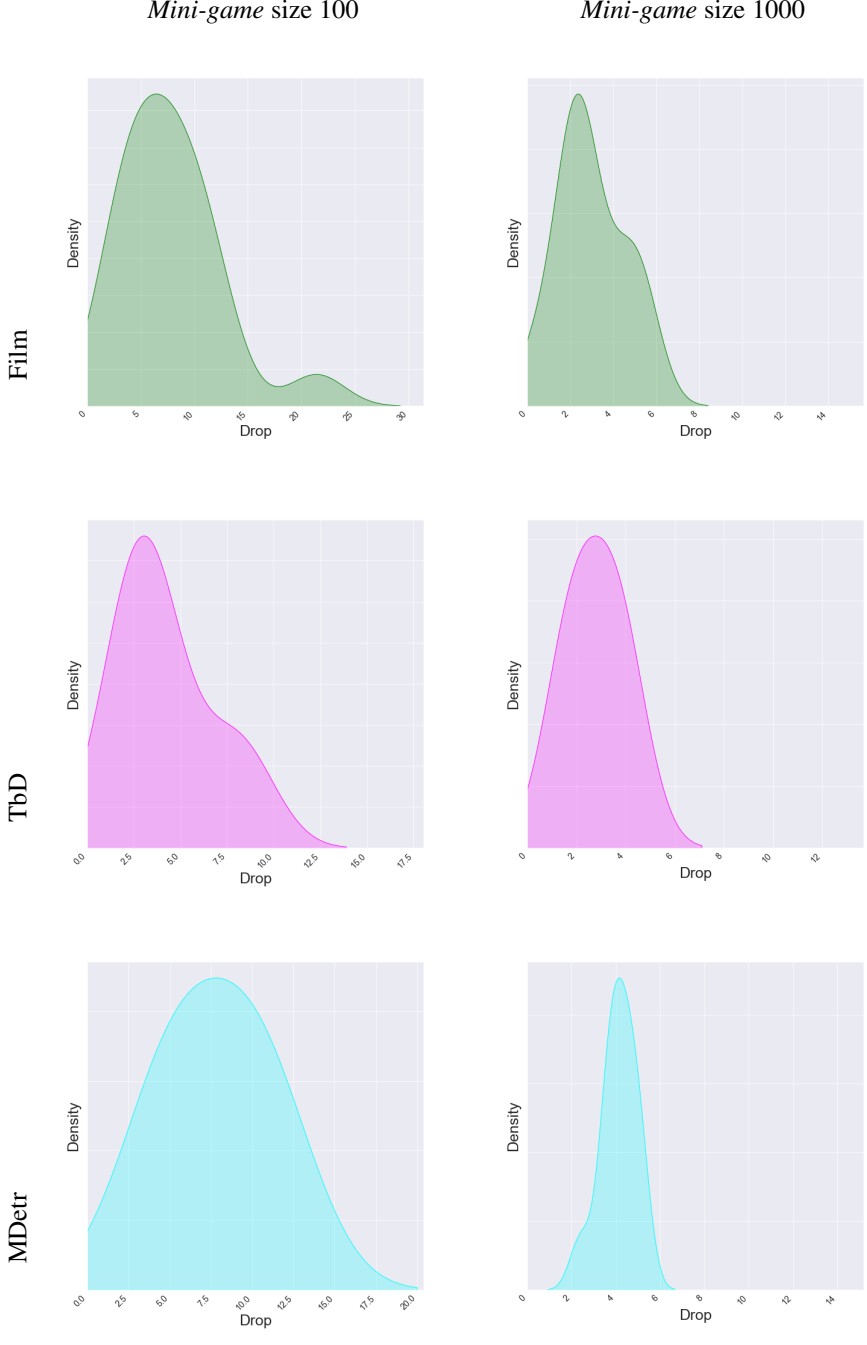

Figure 10: *Mini-games* performance distribution for *state-input Adversarial Player*.

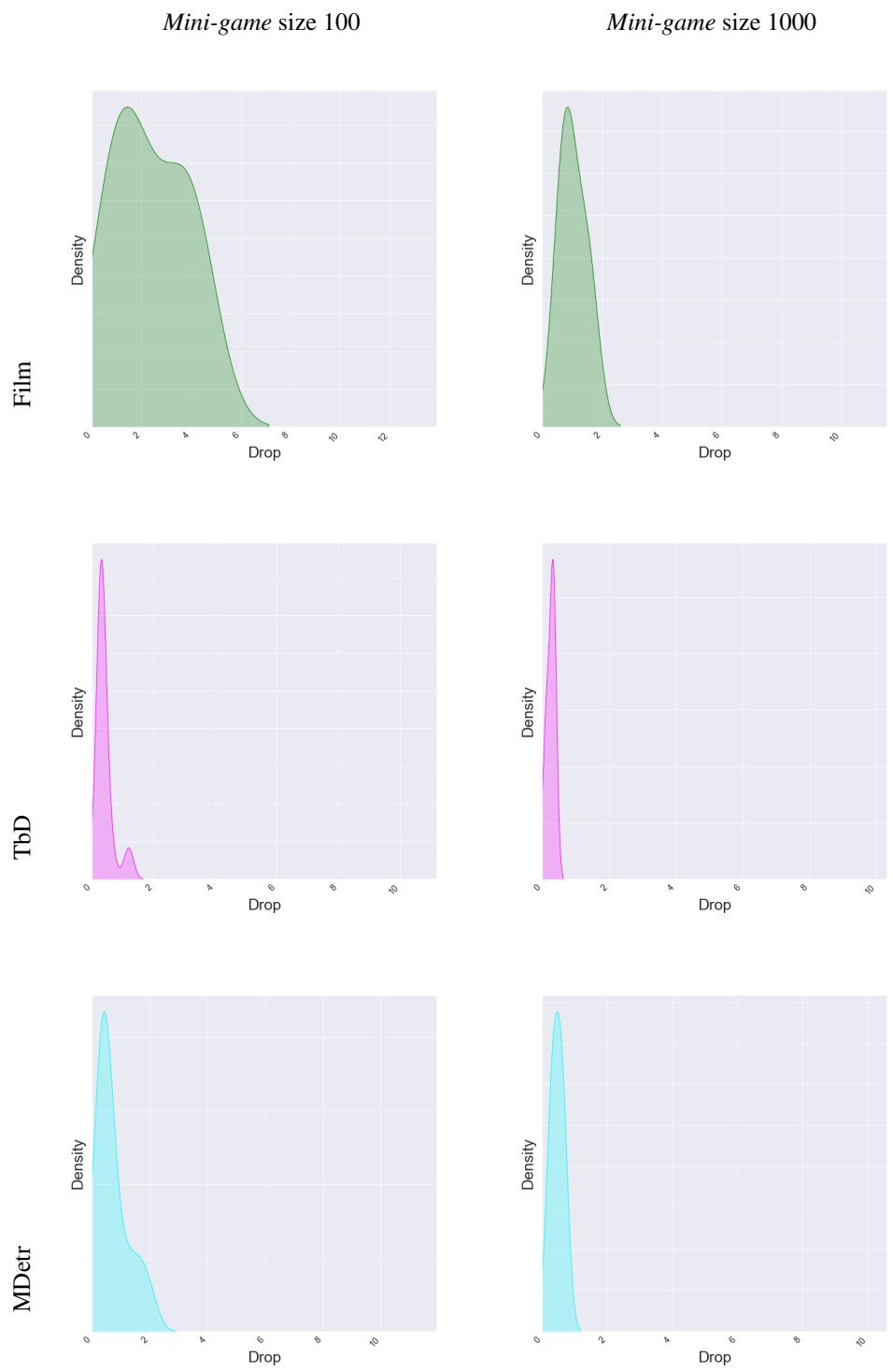

Figure 11: *Mini-games* performance distribution for *pixel-input Adversarial Player*.

## A.9 OUT-OF-DISTRIBUTION MANIPULATIONS

If we do not provide extra constraints on the scene generation process with our environment enforcers, *Adversarial Player* may find an easy manipulation that changes the camera pose or location, effectively 'zooming out' the whole scene. It does so by stretching all the object coordinates in the scene. Figure 12 illustrates that. For instance, *large* objects look much smaller than typical examples. As such constructed scenes are quite different from the ones that *Visual-QA Player* has observed during training, we categorize such manipulations to be *out-of-distribution*. Note that, due to our *in-distribution* environment enforcers, such scenes are prohibited in our pipeline.

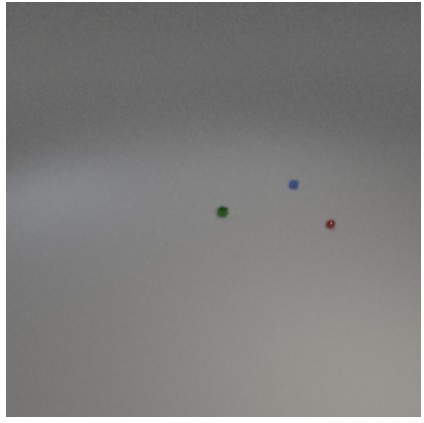
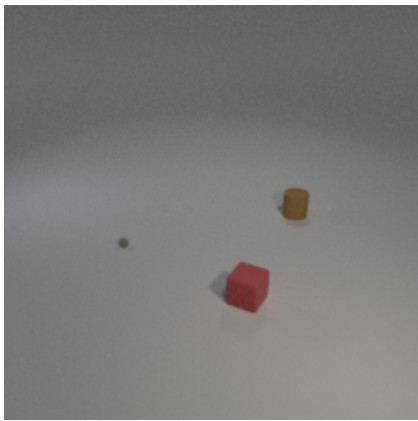

(a) Shape of the blue cube is misclassified as sphere.  (b) Large, brown cylinder is misclassified as small.

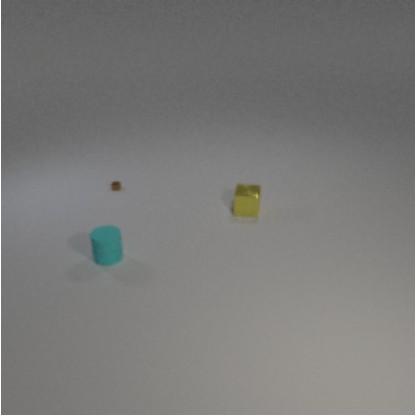

(c) Large, yellow cube is misclassified as small.

Figure 12: Out-of-distribution examples.

### A.10    ADDITIONAL QUALITATIVE RESULTS

In Figures 13-21, we provide more qualitative results.

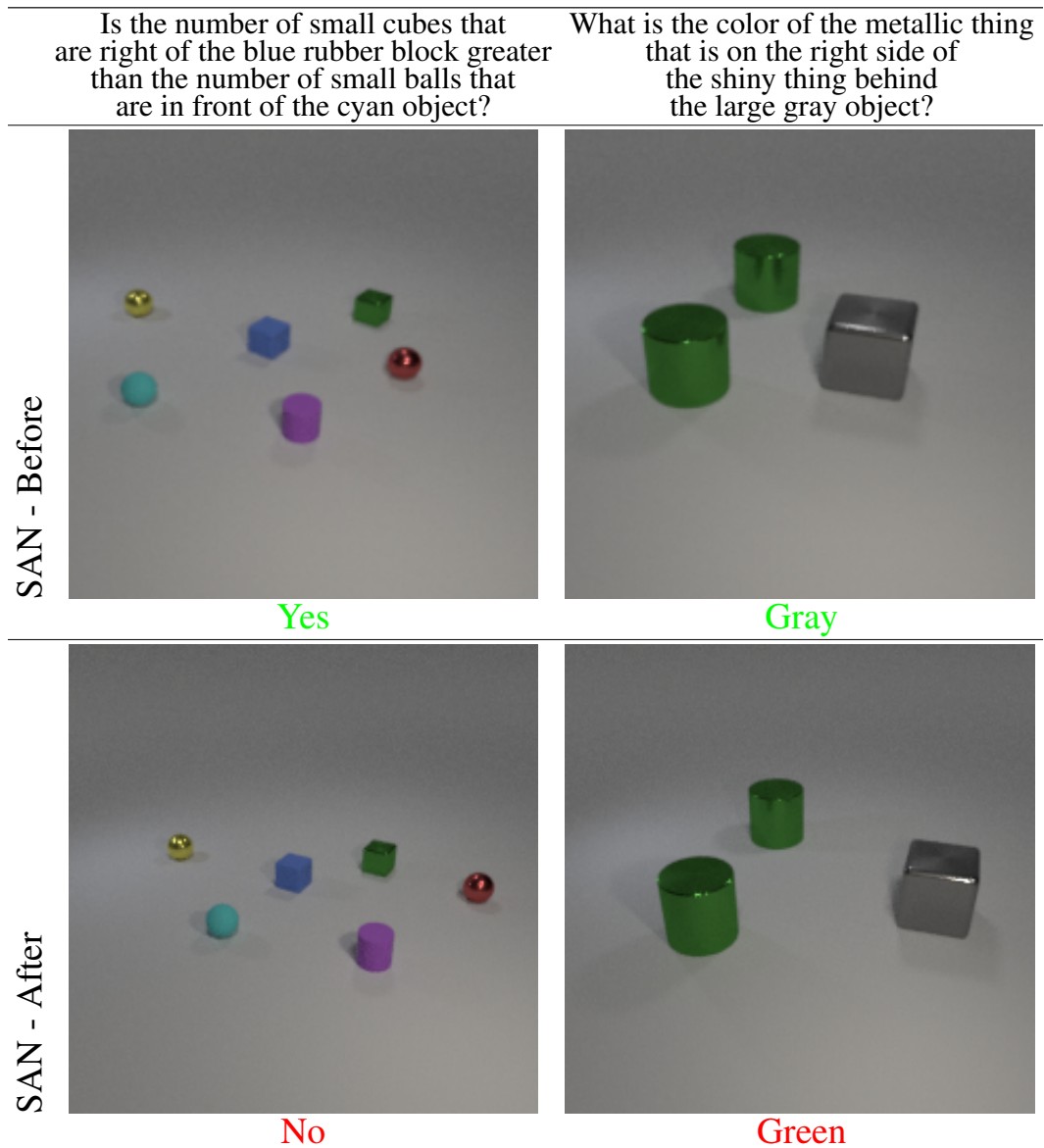

Figure 13: Manipulations of CLEVR models. We show results *before* and *after* scene manipulations.

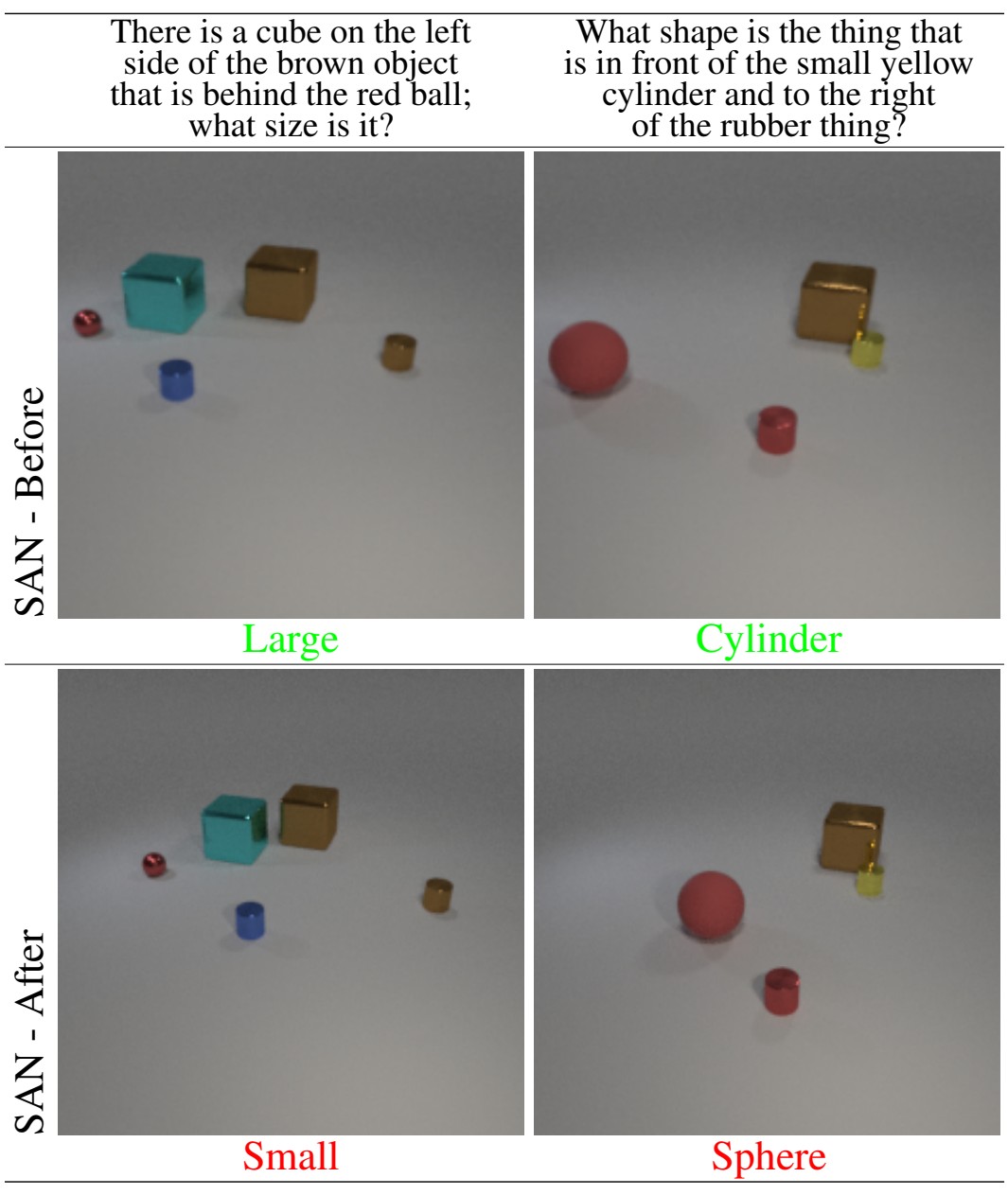

Figure 14: Manipulations of CLEVR models. We show results *before* and *after* scene manipulations.

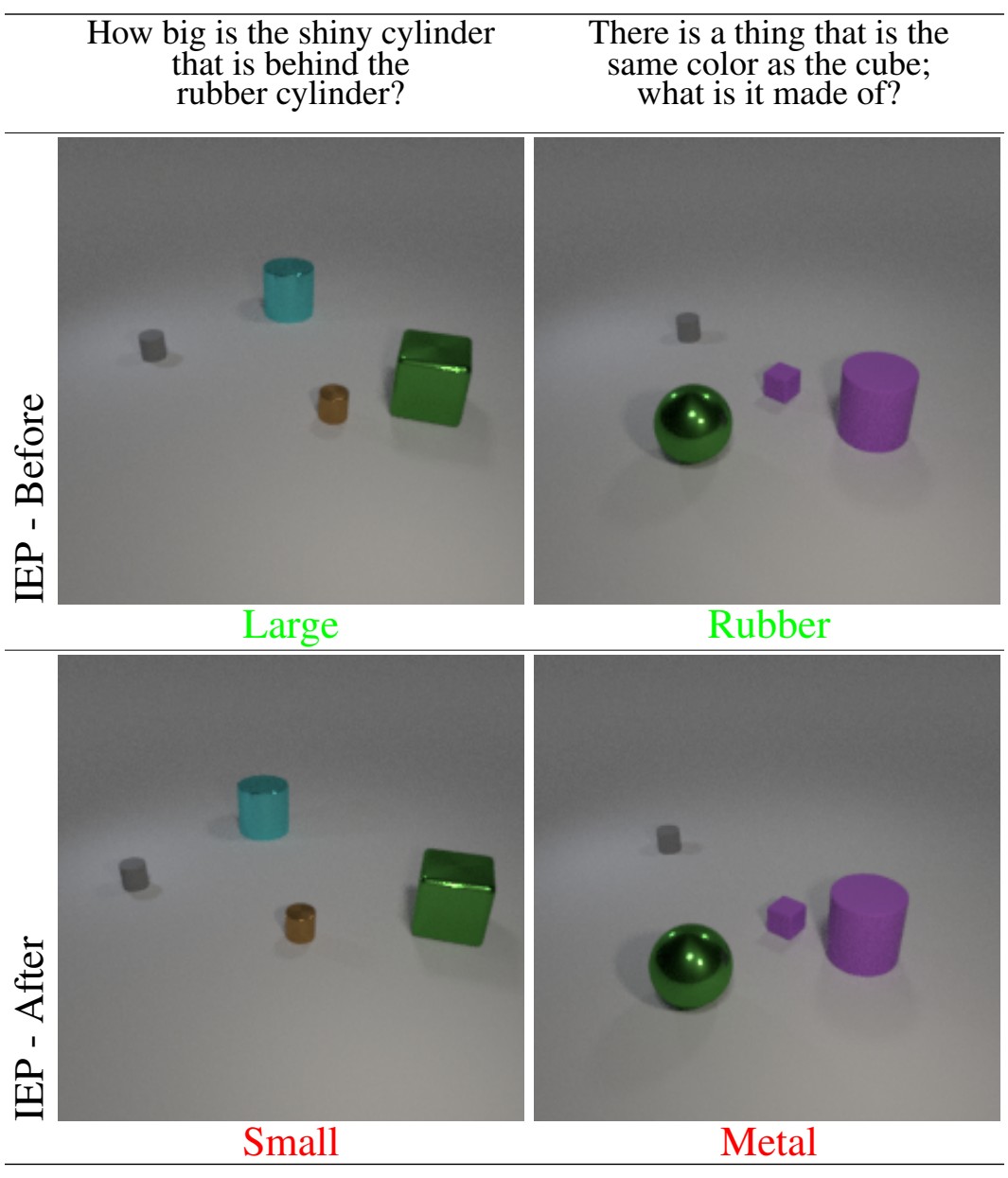

Figure 15: Manipulations of CLEVR models. We show results *before* and *after* scene manipulations.

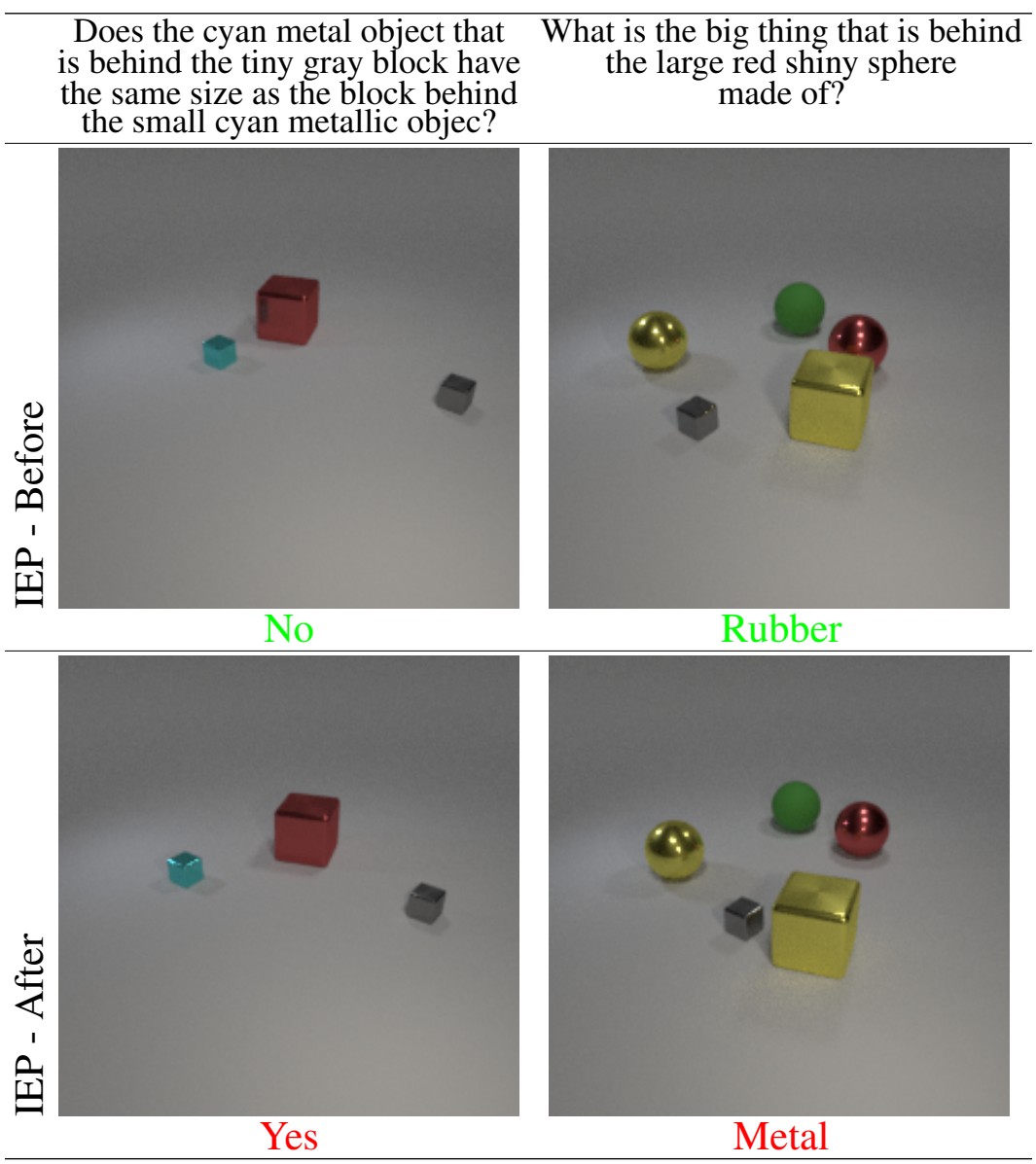

Figure 16: Manipulations of CLEVR models. We show results *before* and *after* scene manipulations.

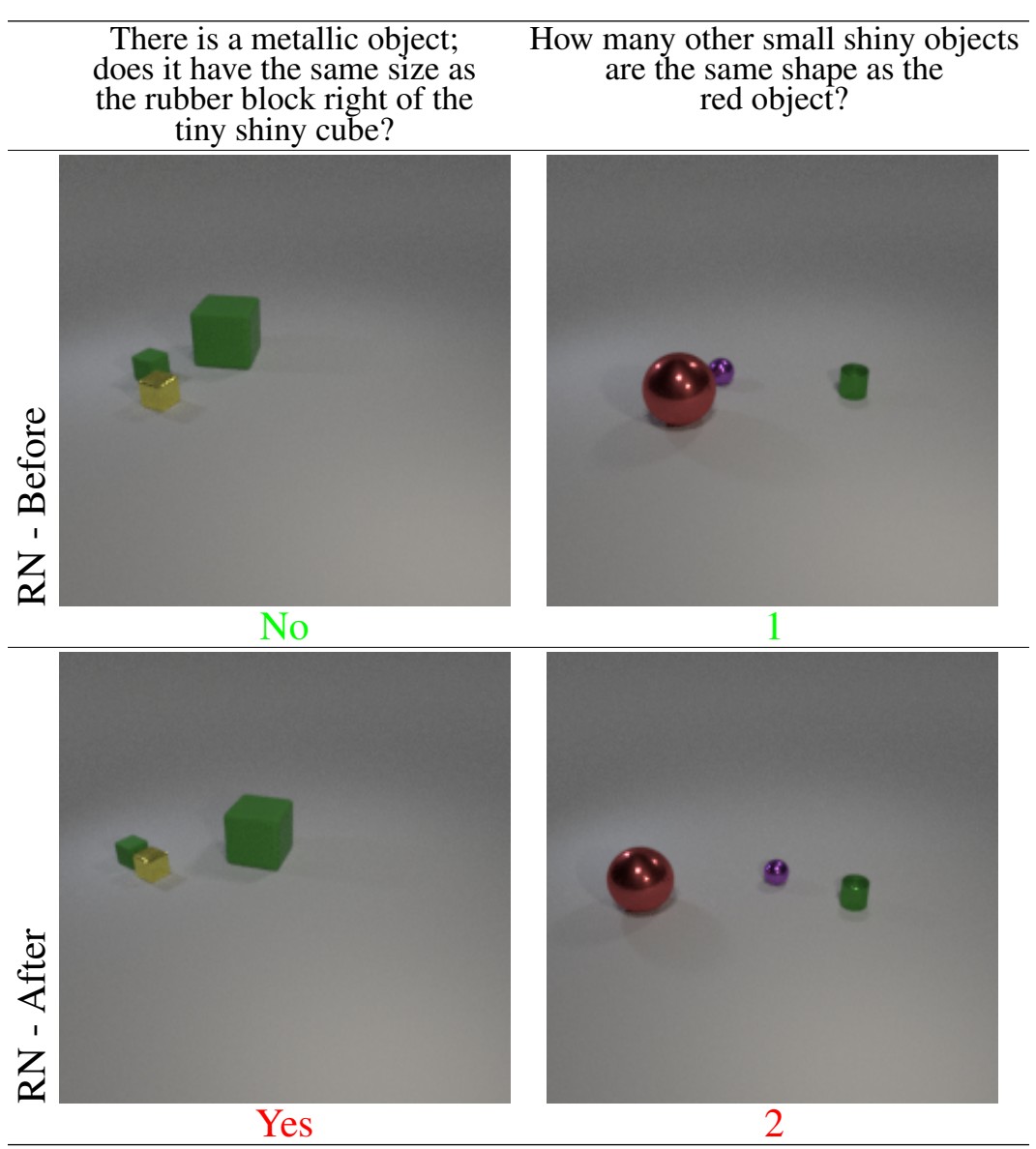

Figure 17: Manipulations of CLEVR models. We show results *before* and *after* scene manipulations.

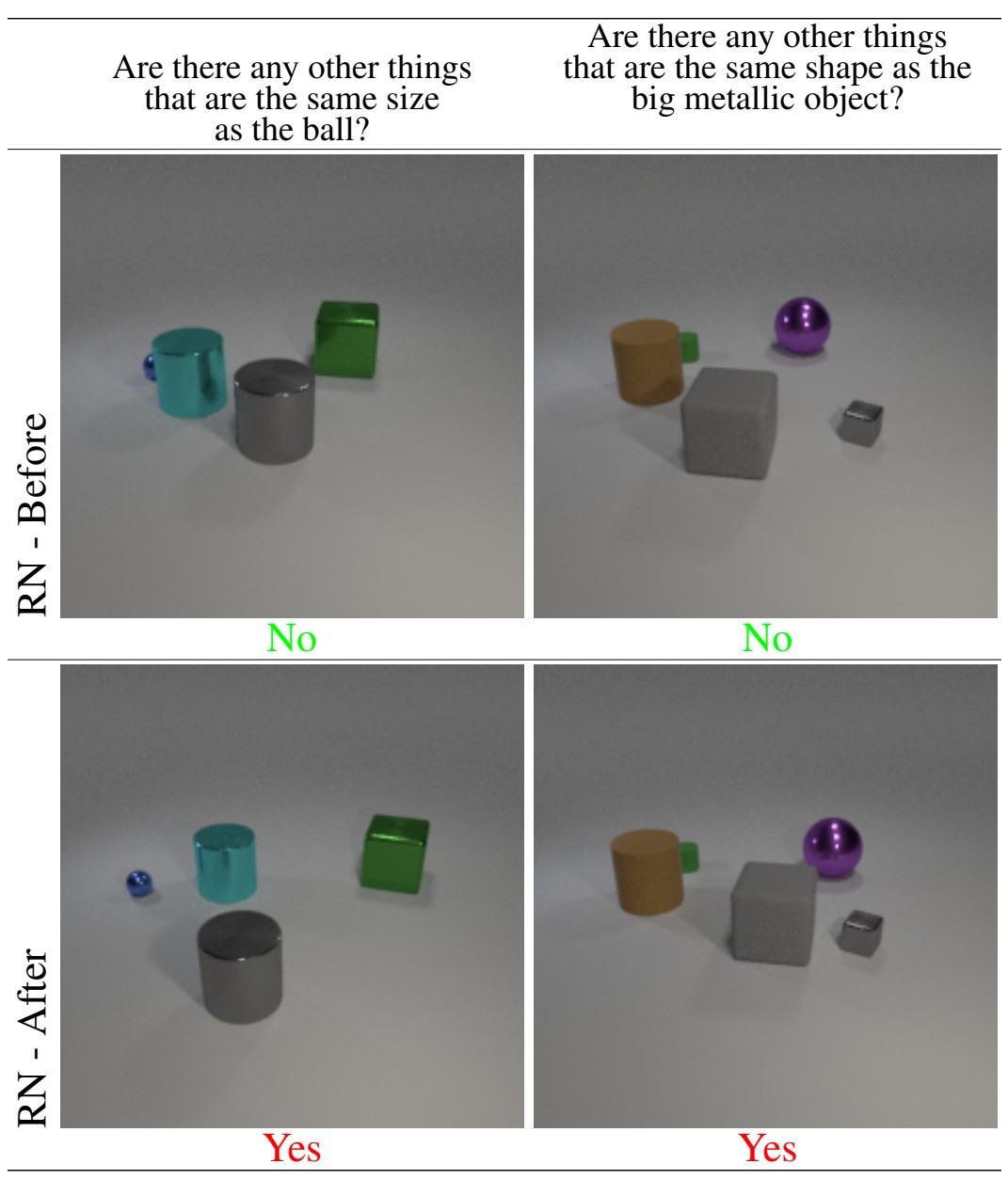

Figure 18: Manipulations of CLEVR models. We show results *before* and *after* scene manipulations.

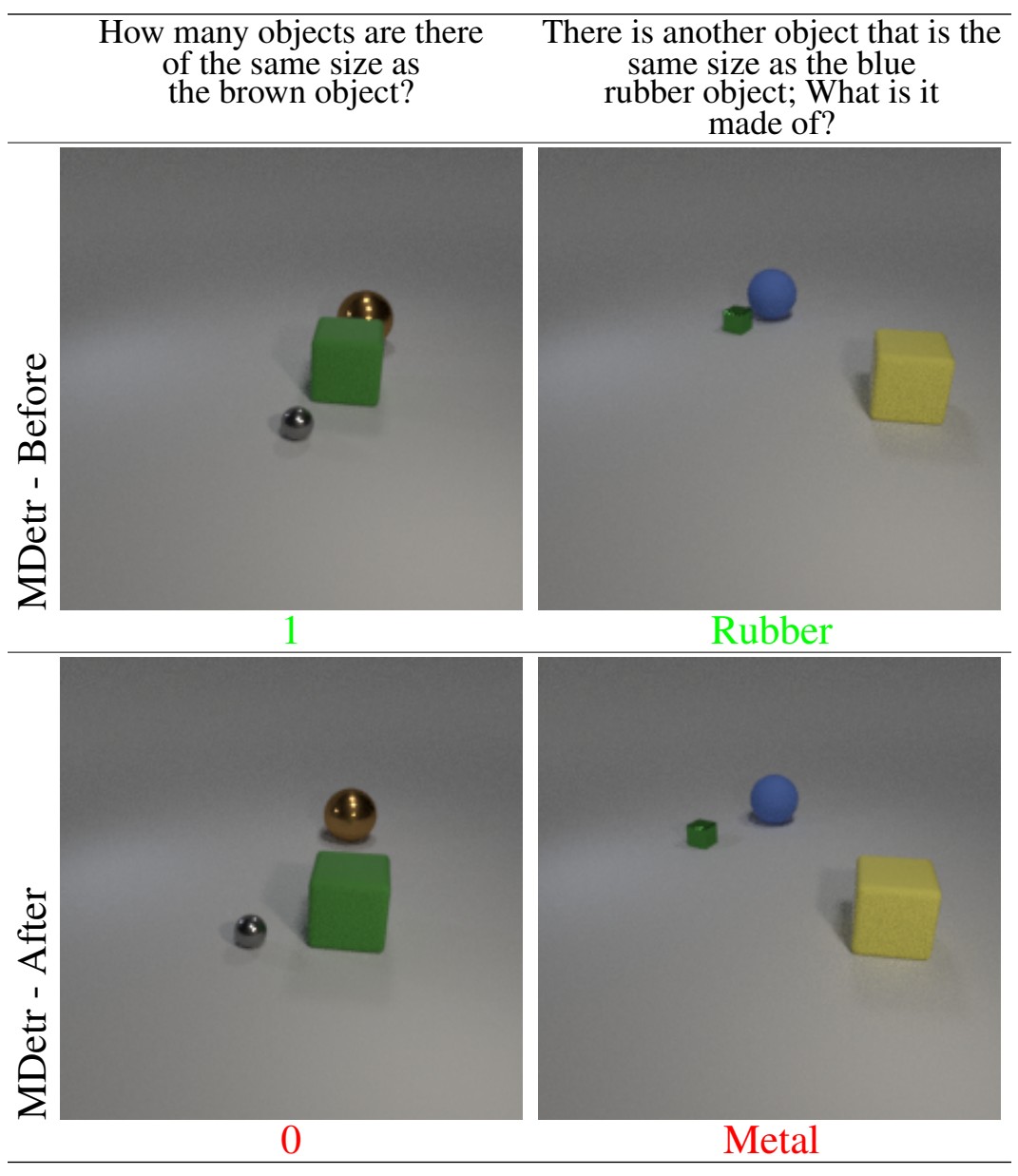

Figure 19: Manipulations of CLEVR models. We show results *before* and *after* scene manipulations.

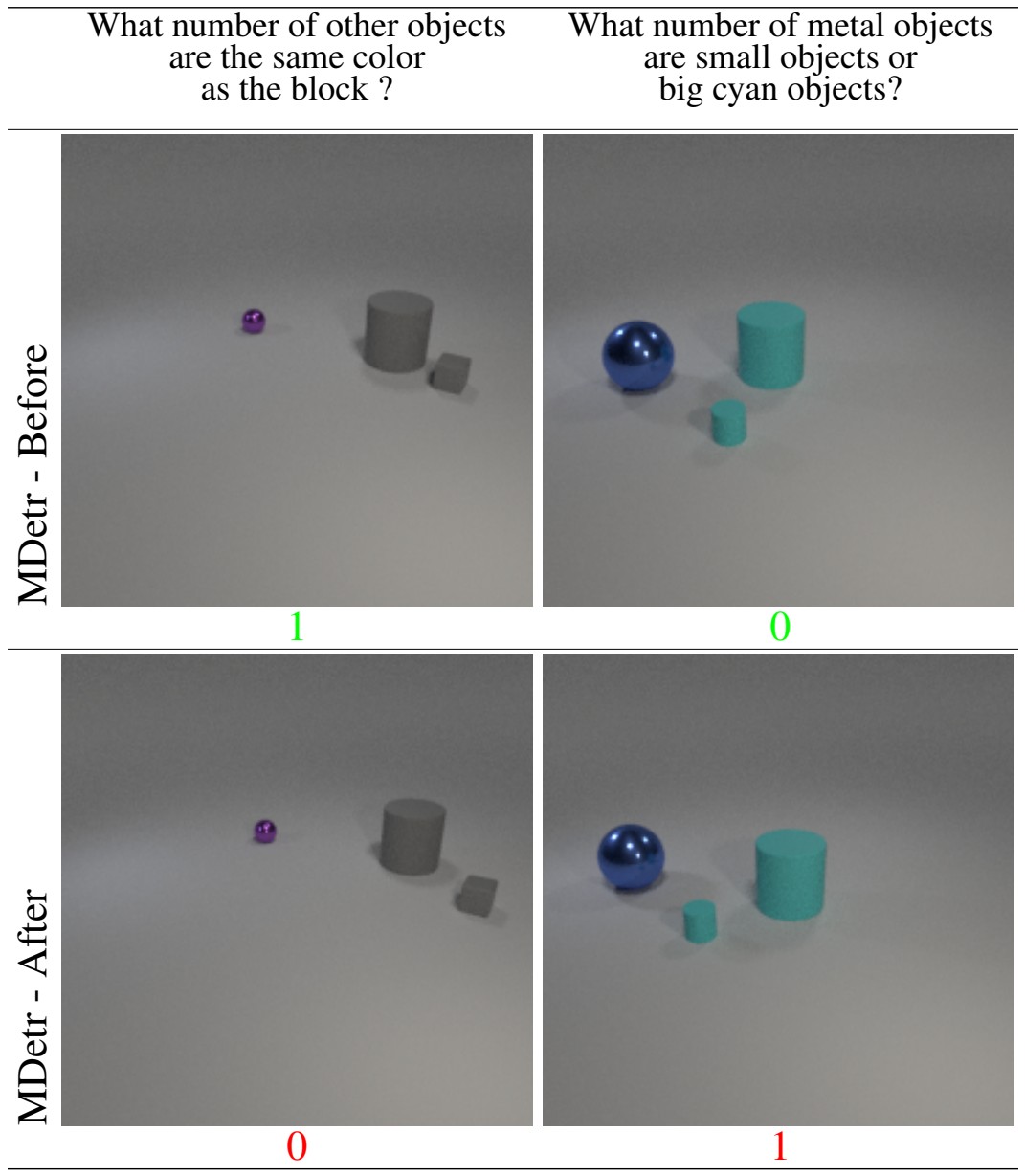

Figure 20: Manipulations of CLEVR models. We show results *before* and *after* scene manipulations.

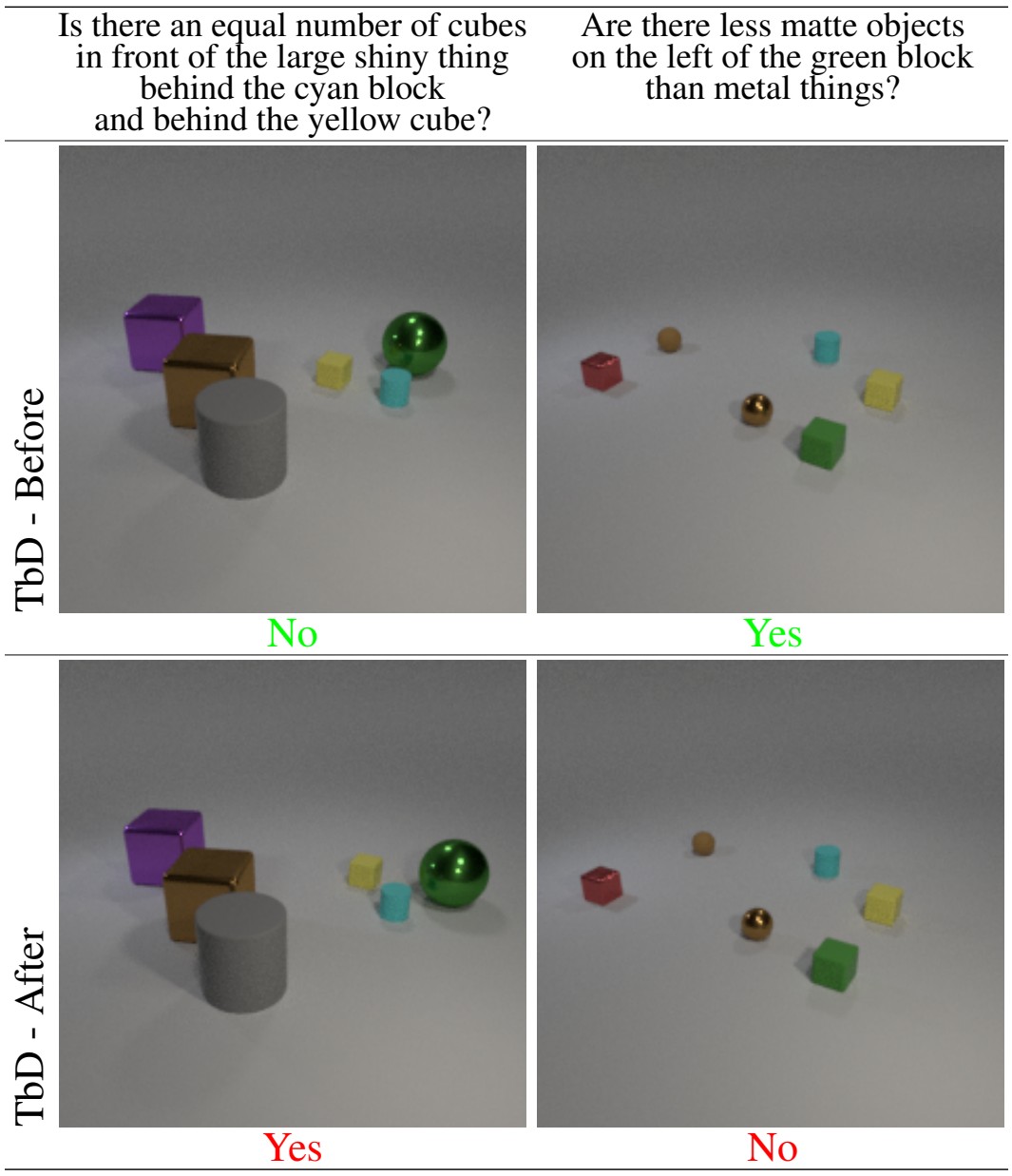

Figure 21: Manipulations of CLEVR models. We show results *before* and *after* scene manipulations.

### A.11 QUANTITATIVE RESULTS WITH p-VALUES

We extend the quantitative results of the main paper by also reporting $p$-values of the one-sample hypothesis testing. We use the *Accuracy Drop* metric that measures the accuracy drop after the *Visual-QA Player* is manipulated, and the *Consistency Drop* that measures how many times the manipulated *Visual-QA Player* changes its answer, independently if that is a correct or wrong answer. The higher *Drop Accuracy* or *Consistency Drop*, the more successful manipulations are. We use T-Test (Helmert, 1876; Lüroth, 1876) for each metric all the runs. Our null hypothesis is that the population mean of all the games is zero, indicating that a manipulation is unsuccessful, and the results of our *Adversarial Players* are extreme cases of a good performance, i.e, $H_0 : \mu_l = 0, l \in [\text{Drop}, \text{Consistency}]$. As an alternative hypothesis, we assume that the population mean is greater than zero, i.e., $H_A : \mu_l > 0, l \in [\text{Drop}, \text{Consistency}]$. The same setup stands for both the *Consistency* and the *Drop* metric. Tables 4 and 5 show the full table with $p$-values computed. Note that only in two cases, *Adversarial Player* does not manipulate the scene convincingly (TbD and Mdetr for the *pixel-input Adversarial Player*).

Table 4: Results of the game with the *state-input Adversarial Player*.

| Model | *Mini-game* size | Average Consistency Drop % | Accuracy Drop % | p-value Consistency Drop | Accuracy Drop |
|---|---|---|---|---|---|
| SAN | 10 | -23.9 | -14.2 | 0.000 | 0.000 |
| | 100 | -10.7 | -8.3 | 0.000 | 0.000 |
| | 1000 | -3.5 | -2.5 | 0.001 | 0.000 |
| FiLM | 10 | -14.8 | -12.7 | 0.003 | 0.006 |
| | 100 | -7.8 | -7.3 | 0.000 | 0.000 |
| | 1000 | -3.1 | -2.4 | 0.000 | 0.000 |
| RN | 10 | -20.9 | -13.6 | 0.000 | 0.000 |
| | 100 | -10.5 | -8.2 | 0.000 | 0.000 |
| | 1000 | -3.3 | -2.8 | 0.000 | 0.000 |
| IEP | 10 | -13.9 | -13.0 | 0.001 | 0.003 |
| | 100 | -6.9 | -6.8 | 0.000 | 0.000 |
| | 1000 | -2.9 | -2.8 | 0.000 | 0.000 |
| TbD | 10 | -5.3 | -5.1 | 0.011 | 0.012 |
| | 100 | 3.8 | -2.5 | 0.000 | 0.000 |
| | 1000 | -1.1 | -1.1 | 0.000 | 0.000 |
| Mdetr | 10 | -6.6 | -6.0 | 0.018 | 0.021 |
| | 100 | -4.9 | -3.6 | 0.000 | 0.000 |
| | 1000 | -1.3 | -1.2 | 0.000 | 0.000 |
| State Input Model | 10 | -8.5 | -7.7 | 0.020 | 0.034 |
| | 100 | -2.3 | -2.1 | 0.000 | 0.000 |
| | 1000 | -1.2 | -1.1 | 0.000 | 0.000 |

Table 5: Results of the game with the *pixel-input Adversarial Player*.

| Model | Mini-game size | Average | | p-value | |
| --- | --- | --- | --- | --- | --- |
| | | Consistency Drop % | Accuracy Drop % | Consistency Drop | Accuracy Drop |
| SAN | 10 | -12.8 | -11.4 | 0.008 | 0.018 |
| | 100 | -6.4 | -3.9 | 0.000 | 0.001 |
| | 1000 | -2.3 | -1.4 | 0.000 | 0.000 |
| FiLM | 10 | -3.62 | -2.6 | 0.056 | 0.154 |
| | 100 | -1.7 | -1.4 | 0.000 | 0.000 |
| | 1000 | -0.6 | -0.4 | 0.000 | 0.000 |
| RN | 10 | -9.4 | -7.2 | 0.021 | 0.065 |
| | 100 | -3.2 | -2.5 | 0.000 | 0.000 |
| | 1000 | -2.3 | -1.7 | 0.000 | 0.000 |
| IEP | 10 | -2.9 | -2.5 | 0.111 | 0.171 |
| | 100 | -1.5 | -1.3 | 0.001 | 0.004 |
| | 1000 | -0.4 | -0.3 | 0.000 | 0.000 |
| TbD | 10 | 0.0 | 0.0 | 1.000 | 1.000 |
| | 100 | -0.6 | -0.4 | 0.017 | 0.078 |
| | 1000 | -0.2 | -0.1 | 0.000 | 0.000 |
| Mdetr | 10 | 0.0 | 0.0 | 1.000 | 1.000 |
| | 100 | -1.2 | -1.1 | 0.002 | 0.003 |
| | 1000 | -0.4 | -0.3 | 0.000 | 0.000 |
| State | 10 | -0.9 | -0.8 | 0.033 | 0.038 |
| Input | 100 | -0.7 | -0.7 | 0.031 | 0.031 |
| Model | 1000 | -0.3 | -0.3 | 0.000 | 0.000 |

# B    APPENDIX B

## B.1    MODEL PERFORMANCE ON MAXIMAL DROP MINIGAMES.

In the table below, we present model accuracy on each of the *Mini-games* where the maximal performance drop was observed. The third and fourth columns (Original Data section) refer to the model accuracy on unmanipulated image-question pairs of the respective *Mini-game*. The fifth and sixth columns (Manipulated Data section) refer to the model accuracy in those *Mini-game* examples after the *Adversarial Player* manipulation.

| Model | *Mini-game* size | Original Data | | Manipulated Data | |
|---|---|---|---|---|---|
| | | *Mini-game* Accuracy (State) | *Mini-game* Accuracy (Pixel) | *Mini-game* Accuracy (State) | *Mini-game* Accuracy (Pixel) |
| SAN | 10 | 80.00 | 80.00 | 28.84 | 43.26 |
| | 100 | 74.00 | 75.00 | 47.65 | 66.32 |
| | 1000 | 72.30 | 72.20 | 68.42 | 70.87 |
| FiLM | 10 | 100.0 | 100.0 | 48.10 | 86.58 |
| | 100 | 98.00 | 100.0 | 75.61 | 92.44 |
| | 1000 | 96.40 | 96.10 | 90.81 | 94.75 |
| RN | 10 | 100.0 | 100.0 | 46.60 | 74.56 |
| | 100 | 94.00 | 95.00 | 63.28 | 87.42 |
| | 1000 | 93.10 | 93.30 | 90.03 | 91.05 |
| IEP | 10 | 100.0 | 100.0 | 48.45 | 87.21 |
| | 100 | 98.00 | 96.00 | 74.41 | 93.50 |
| | 1000 | 97.10 | 97.30 | 93.99 | 95.73 |
| TbD | 10 | 100.0 | 100.0 | 69.37 | 99.10 |
| | 100 | 100.0 | 100.0 | 91.17 | 98.00 |
| | 1000 | 99.40 | 99.70 | 95.53 | 98.80 |
| Mdetr | 10 | 100.0 | 100.0 | 59.82 | 99.70 |
| | 100 | 100.0 | 100.0 | 86.43 | 97.80 |
| | 1000 | 99.50 | 100.0 | 94.51 | 99.10 |
| State | 10 | 100.0 | 100.0 | 77.44 | 91.96 |
| Input | 100 | 97.00 | 99.00 | 92.63 | 95.05 |
| Transf. | 1000 | 97.20 | 96.40 | 95.15 | 95.83 |

## B.2    GENERATING 'VISUALLY FAIR' SCENES.

We define a scene as *visually fair* if there are no occlusions and all objects are within the field-of-view. The ground truth answer of a posed question on a *visually fair* scene remains the same before and after object manipulation.

For the generation of new images/scenes we use the open-source Blender Graphics Engine [2] (v2.79b), and the original 3D models of the CLEVR dataset. Scenes suggested by our *Adversarial Player*, are encoded in the form of arguments for Blender, which renders the new images according to those directives. In order to guarantee visual fairness, and question validity, we employ two modules (scene-constraint enforcer / question relevance enforcer), presented briefly - in Section 4 in the Consistency and In-Distribution paragraphs.

The scene-constraint enforcer is responsible for ensuring the creation of a scene/image that respects all constraints and statistical properties of the original dataset. This piece of code is borrowed from the original CLEVR codebase [3] and as in the original CLEVR paper the following constraints are checked (Lines 66-83 in the python code):

---

[2]Blender - a 3D modeling and rendering package. `https://www.blender.org`
[3]`https://github.com/facebookresearch/clevr-dataset-gen/blob/main/image_generation/render_images.py`

1. **Minimum / Maximum Number of Objects:** 3 / 10.

2. **Minimum allowed distance between object centers:** 0.25

3. **Margin along with cardinal directions:** 0.4 (This ensures lack of ambiguity by keeping objects at least 0.4 points of distance apart. This distance is a Blender-specific measure that spans from -3 to +3.)

4. **Min pixels per object:** 200 ( This ensures that an object will not be occluded, by requiring at least 200 pixels of it being visible.)

The question-relevance enforcer checks if the newly generated scene is indeed answerable and has the same "ground truth" answer as the old scene. By not enforcing this, every manipulation will be falsely "successful" as the question (which is kept the same before and after the attack) will have a different answer in the newly generated scene.

This is also resolved by using the same question engine module provided by the CLEVR authors in their codebase[4]. The module receives a question and a scene and calculates the ground truth answer based on a set of functional programs that are apriori known and used to generate the questions. (Supplementary Material A in the CLEVR paper (Johnson et al., 2017a))

We use the following pipeline. Our *Adversarial Player* initially receives an input image-question pair. Then, it suggests a new scene, that passes through the scene-constraint enforcer. If the new scene respects all boundaries and constraints it is then evaluated alongside the question by the question-relevance enforcer. If the new ground truth answer to the suggested scene/image is the same as the original answer, the image-question pair is marked as in-distribution and valid. Otherwise, it is removed from the training pipeline.

---

[4]https://github.com/facebookresearch/clevr-dataset-gen/blob/main/question_generation/question_engine.py

### B.3 COMPARISON WITH RANDOMLY SAMPLED EQUIVALENT SCENES.

Arguably, out of the semantically equivalent space of images given a question and an accompanying answer, the *Adversarial Player* effectively discovers cases for which the model under the test can be fooled. This, however, does not indicate how rare those examples are, neither the degree of robustness of the model against other examples in this space. Let us limit ourselves on a specific image-question pair. Then, a tractable estimate of how rare "fooling examples" are would be the number of manipulated scenes ($F$) that are able to fool the model, divided by all the possible permutations of available objects ($N$) in our discrete 7x7 grid. $Rarity = \frac{F}{49^N}, N \in [3, 10]$. A scene of only 3 objects and 1 discovered successful manipulation would make it 0.00085% possible to find it by chance and would indicate 99.9915% robustness against that sphere of equivalent image-question pairs.

Of course not all of those configurations are valid due to scene constraint violations, as well as objects can theoretically move freely on the scene. As a way to approach this question, we compare our *Adversarial Player* against a random scene generator (RSG). The random scene generator is able to manipulate any object freely, under the constraint that the resulting scene is valid under the scene constraint enforcer as well as the question relevance enforcer. Manipulations of lighting or camera angles, as well as addition or removal of objects is not permitted. In order to reduce the huge rendering times needed for each random search, we allow the RSG to operate on a 10 length *Mini-game* where scene manipulations that cause accuracy drop have been discovered by the *Adversarial Player*. For each image-question pair the RSG has a budget of 5000 queries, and if not successful it proceeds to the next *Mini-game* entry.

| Model | CLEVR Accuracy | Adv.Player Accuracy (Drop %) | RSG Accuracy (Drop %) |
|-------|----------------|------------------------------|------------------------|
| SAN   | 72.1           | 61.86 (-14.2%)               | 71.59 (-0.7%)          |
| FiLM  | 96.2           | 83.98 (-12.7%)               | 95.81 (-0.4%)          |
| RN    | 93.2           | 80.52 (-13.6%)               | 92.73 (-0.5%)          |
| IEP   | 96.9           | 84.30 (-13.0%)               | 96.70 (-0.2%)          |
| TbD   | 99.1           | 94.04 (-5.1%)                | 99.10 (-0.0%)          |
| MDetr | 99.7           | 93.71 (-6.0%)                | 99.70 (-0.0%)          |
| ST    | 96.8           | 89.34 (-7.7%)                | 96.50 (-0.3%)          |

As it can be seen, randomly question-uninformed suggested object placements do not seem to be effective into fooling the models under test. This further supports the argument that models under test theoretically and statistically seem robust, but nevertheless have vulnerabilities that can be effectively exploited by our *Adversarial Player*.

## B.4 VISUALIZATIONS.

Here, we try to isolate reasoning gaps from visual causes after the scene manipulations; with the following setup. Each *Visual-QA Player* is introduced with two image-question pairs. The first consists of the original image and the second of a scene created by our *Adversarial Player* that successfully fools the *Visual-QA Player*. We then inspect the visualization of the *Visual-QA Player*'s inner-workings in both pairs. We are especially interested in patterns that suggest the root of failure. As each *Visual-QA Player* operates under different assumptions and architectural designs, we choose to visualize :

1. The *attention maps* of Stack-Attention Networks.
2. The *gradient-weighted activations of the pre-ultimate and ultimate convolutional blocks* of FiLM.
3. The *programs* that are synthesized alongside their *soft-attention maps* of each module block in TbD network.
4. The *objects* that were detected during the computation steps of a MDetr model.

In Figure 22, we visualize the pre-ultimate (column 2) and ultimate (column 3) attention maps of the Stack-Attention Network. We observe that the model focuses on a single final object in manipulated examples. In the original ones, the relational jumps required for the answer are still visible on the attention map. The final attention map that is presented (column 3) is passed to a fully-connected classifier in order for the answer to be created. We observe limited focus to all the necessary objects / locations that creates a lack of necessary features for the correct answer derivation.

In the case of FiLM model in Figure 23, we observe a similar scenario. In the first example, (rows 1 and 2) the gradient-weighted activations of the model seem to be focusing on multiple irrelevant objects regarding the final answer, while in the second example (rows 3 and 4) the model seems to include the purple cube mistakenly as one of the possible spheres of the image. In both cases above it is difficult to exactly pin down the root of (reasoning or visual) failures.

MDetr model can be seen in Figure 24 and Figure 25. Here, a specific pattern could not be identified. All objects were correctly classified. It suggests that the cause of failures stem rather from the reasoning steps.

Tbd is presented in Figure 26. In both the original and manipulated scenes the programs that the model generates are exactly the same. This is to no surprise since the program generator module is using exclusively the question as input. Nevertheless, a similar pattern of scattered model focus is presented here as well. The attention maps in both examples are the same between the original and manipulated scenes, in all but the final reasoning step. In the first example (rows 1 and 2) the attention is leaked towards the gray cube that has nothing to do with the correct answer, while in the second (rows 3 and 4) the attention is spread among many matte objects but peaks near the big yellow ball. This is wrong since the word "other" that exists in the question should drive the model focus exclusively to other objects apart from the starting point of the reasoning chain.

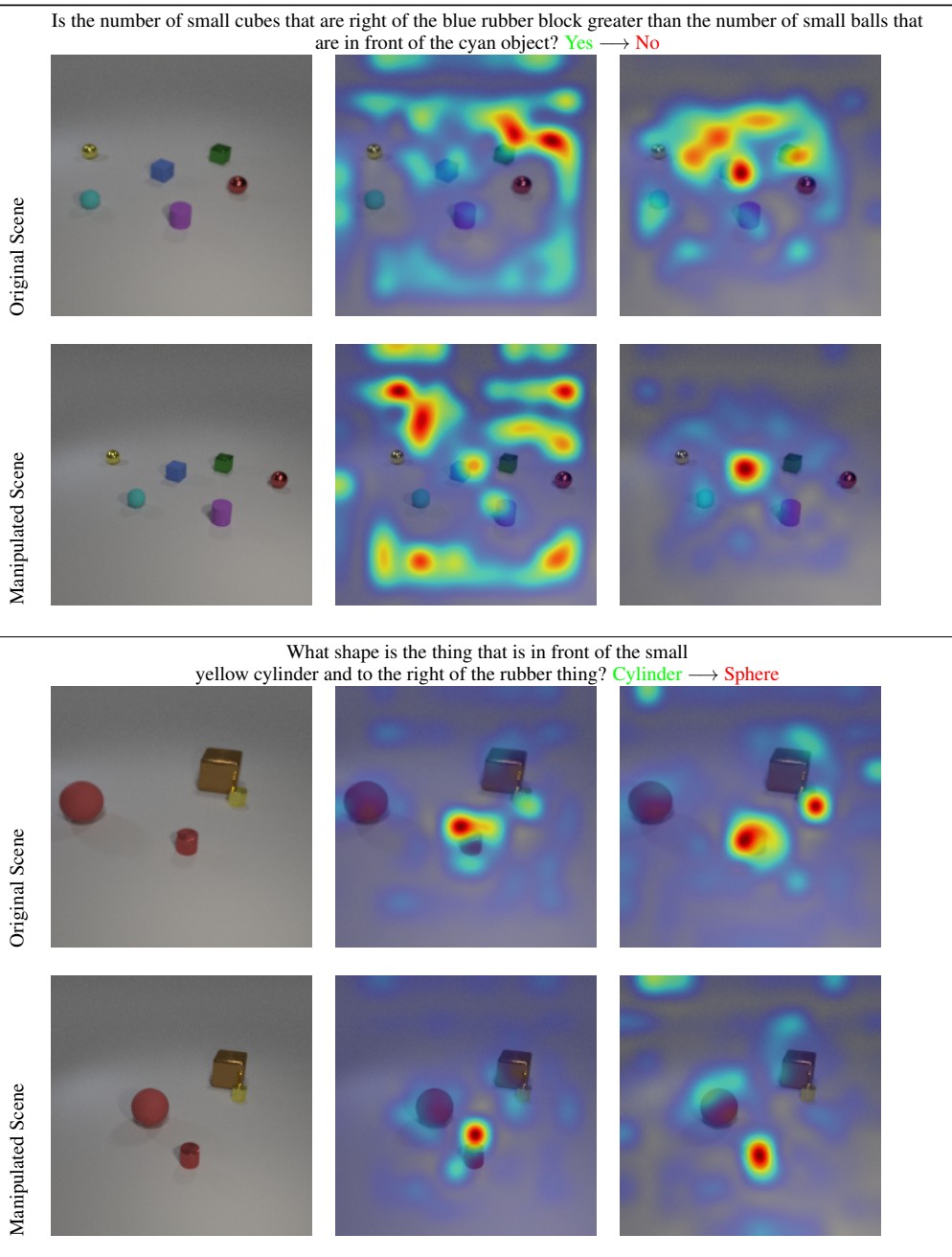

Figure 22: Visualization of attention maps in SAN model on original and manipulated scenes.

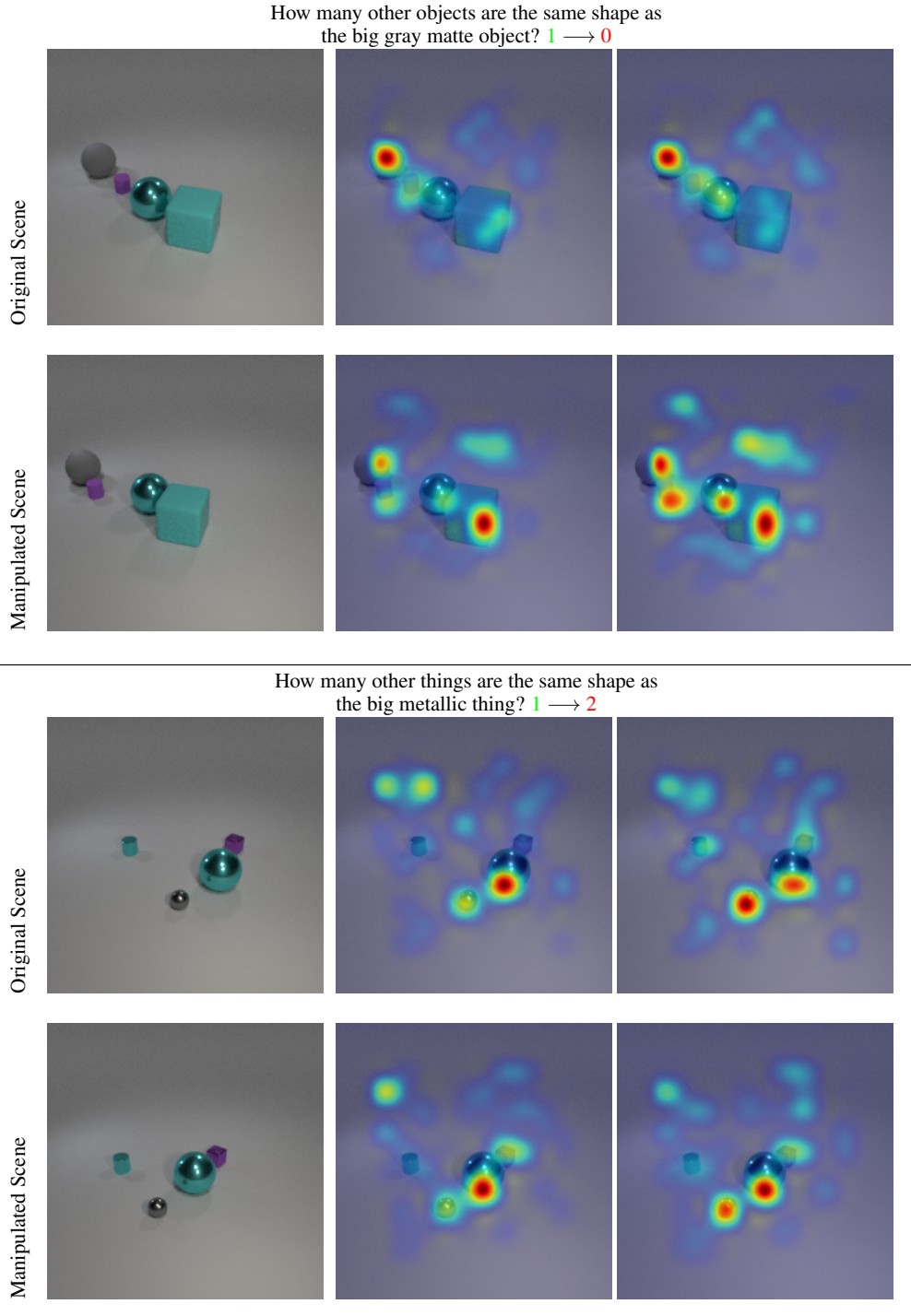

Figure 23: Gradient weighted activation visualization of FiLM model's pre-ultimate and ultimate blocks on original and manipulated scenes

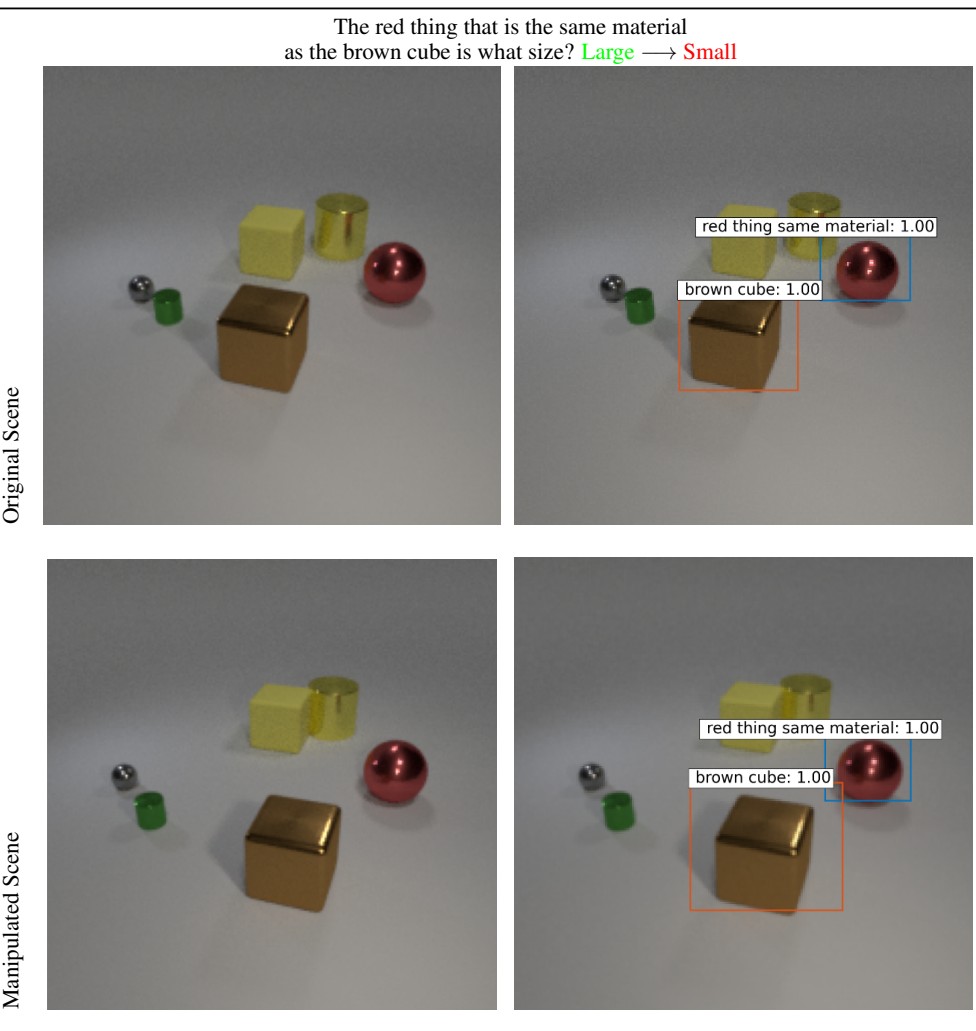

Figure 24: Visualization of MDetr model's detection stages on original and manipulated scenes.

Do the object behind the big matte sphere and the
thing on the right side of the red thing
have the same material? Yes ⟶ No

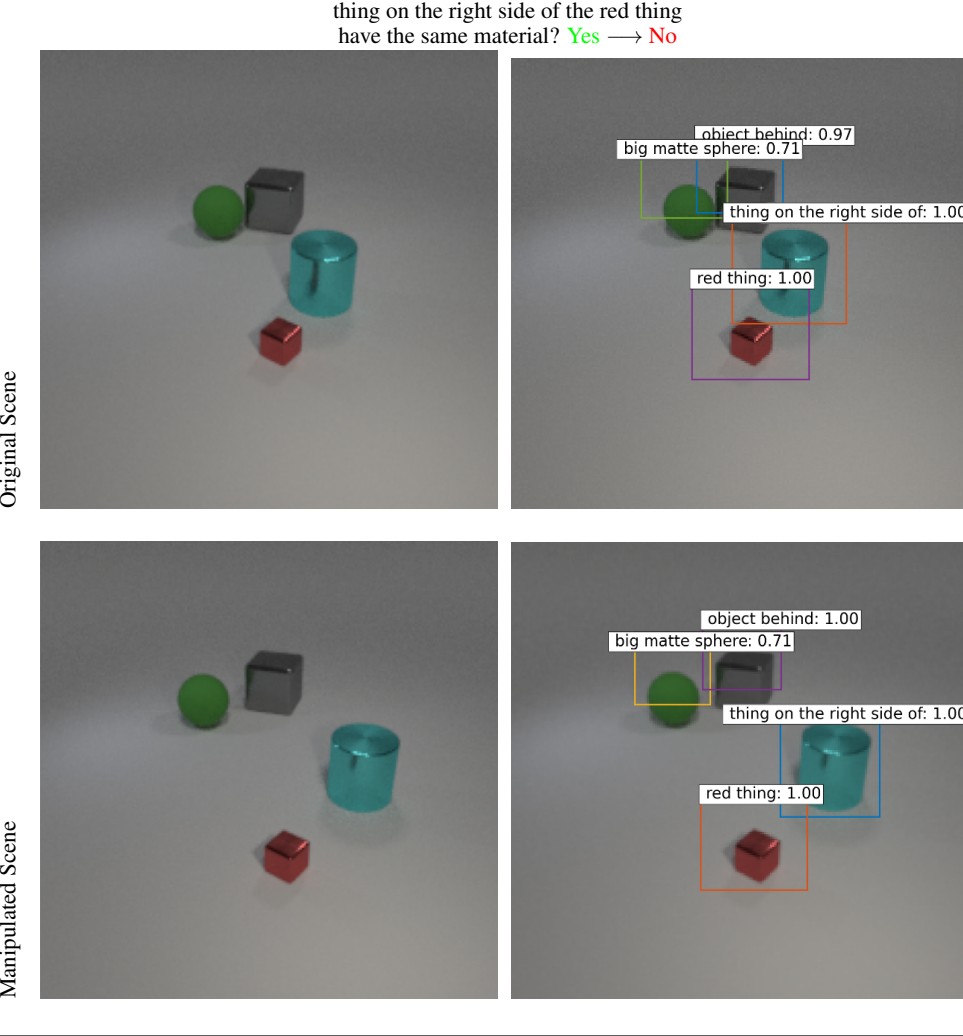

Figure 25: Visualization of MDetr model's detection stages on original and manipulated scenes.

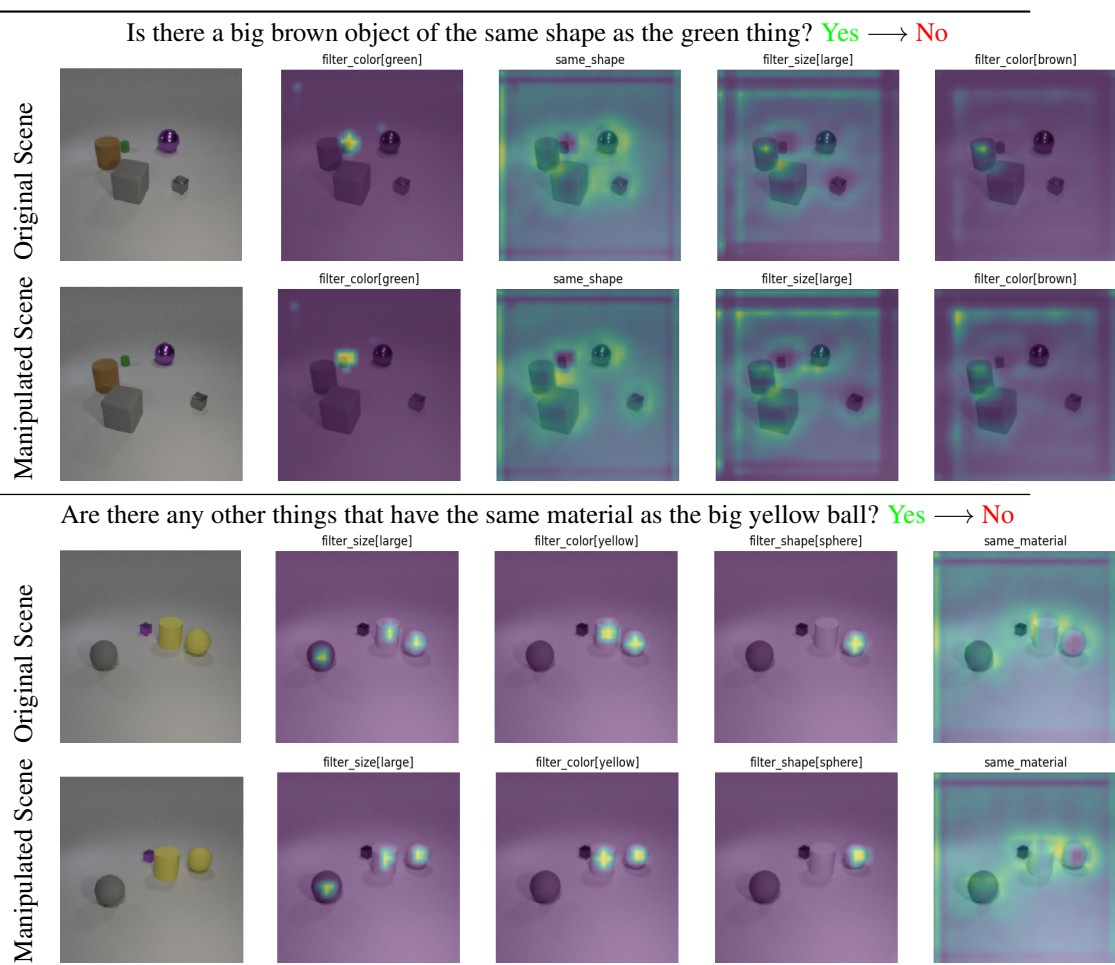

Figure 26: Visualization of TbD model's program generation and attention map visualization on original and manipulated scenes

### B.5 Human-in-the-loop Experiment

Can our *Adversarial Player* 'fool' a human-agent? The perceptual abilities of human observers have been previously measured (92.6%) on this dataset, however, human robustness is a different question. We, hence, decided to assess human robustness by a simple experiment. In our setting, a human observer is presented with a set of two images and a question. Then the observer is required to answer to each image question pair, (image before - question / image after - question), and report whether the answer has changed or remained the same. Participants were first made familiar with the concepts of the CLEVR dataset (colors / shapes / materials / sizes) and then were left unsupervised to answer a set of 20 such examples. Each contained one CLEVR question and two CLEVR scenes, before and after the *Adversarial Player* manipulations. Half of those scenes successfully fooled models and half of those did not. An example of what a survey question looks like is given in Figure 27, together with the confusion matrix Figure 28 (results of our survey).

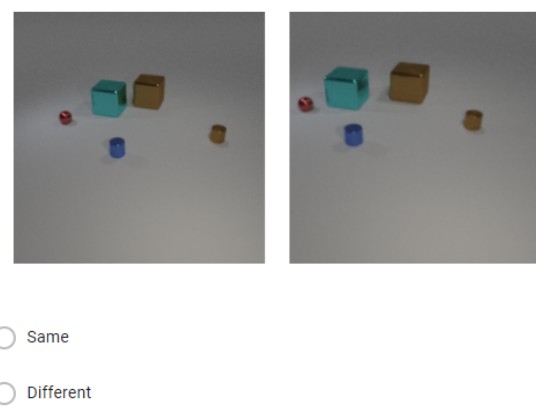

Figure 27: Questionnaire example: The observer is presented with two images and a question. Then they have to pick if their answer would remain the same if the question referred to each image respectively.

As we can see in Figure 28, human observers exhibit 91% precision in identifying cases where scene manipulations are not causing changes to the questions answer, and 95% precision in identifying cases where the manipulations do cause a change. While this small scale experiment may not reflect a universal truth about human robustness in VQA scenarios, it is a suggestion that human observation is still reliable after our agent's manipulations.

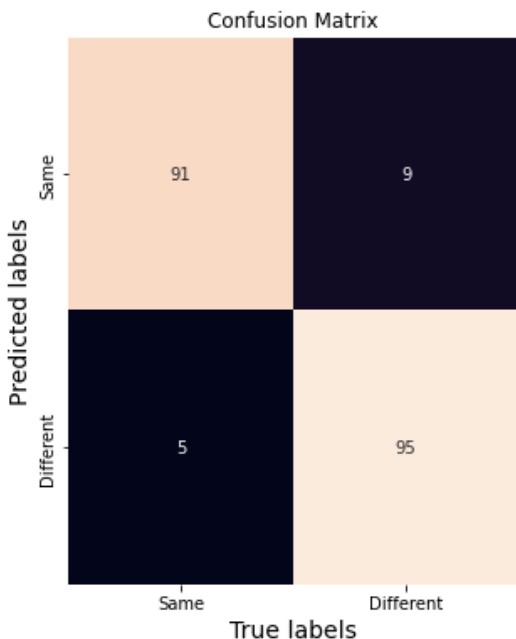

Figure 28: Confusion matrix of human experiment: Participants identified 91% of cases where manipulations were not causing a change in the ground-truth answer. All of these managed to fool *Visual-QA Players* and made up half of the survey. When faced with manipulations that changed the ground-truth answer (the other half of the survey), they identified 95% of them.

## B.6 OBJECT DETECTION EXPERIMENT

As an extra test to confirm the visual fairness (see section B.2) of *Adversarial Player* manipulations we run the following experiment. We employed an unsupervised object discovery model, called Slot Attention (Locatello, 2020), pre-trained on CLEVR, together with a classifier trained to identify object attributes. Next, we used that model to identify objects and attributes on original and manipulated scenes. Ultimately, we could not find any pairs in which the number of objects or their attributes were different before and after the manipulations. The results for two pairs can be seen in Figure 29 and Figure 30. We thus conclude that our *Adversarial Player* manipulations create 'visually fair' scenes, without producing any corner cases of perceptual system.

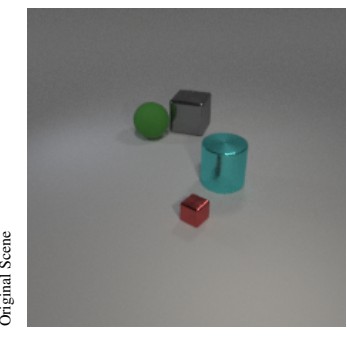

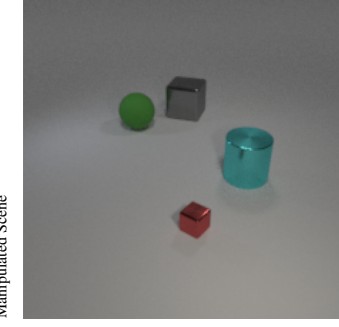

Figure 29: Visualization of Slot Attention model's detection results

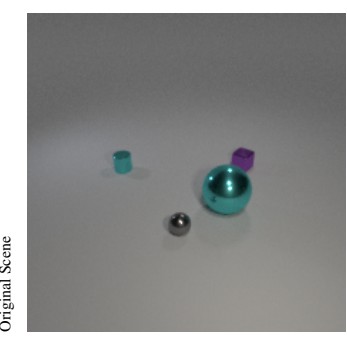

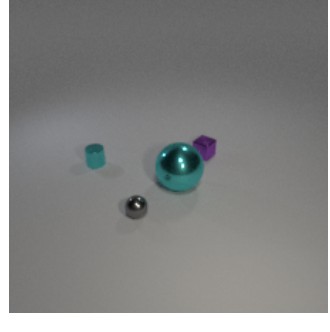

Figure 30: Visualization of Slot Attention model's detection results

