# OpenReview forum: "Measuring CLEVRness: Black-box Testing of Visual Reasoning Models"
_ICLR.cc/2022/Conference — ICLR 2022 Poster_

### Official Review · Reviewer_8VmJ · 2021-11-02

**Correctness:** 3
**Technical Novelty And Significance:** 3
**Empirical Novelty And Significance:** 3
**Recommendation:** 6
**Confidence:** 4

**Main Review:**

Strengths:

The paper proposes a general framework to evaluate the reasoning capabilities of current VQA models. This evaluation could be useful as previous VQA models are likely to overfit the dataset bias instead of understanding and reasoning the question.


Weaknesses:

The experimental results don't reveal much new information. The early models(SAN, FiLM, and RN) are simple and do not consider the reasoning abilities. TbD and Mdetr perform object-level reasoning. Thus it is predictable that they can maintain the average performance given the manipulated samples. Other State-of-the-art models on CLEVR, such as MAC[1] and NS-CL[2], are also likely to have consistent answers. The experimental results do not expose any new weakness of the evaluated VQA methods.

[1] Drew A. Hudson, Christopher D. Manning, Compositional Attention Networks for Machine Reasoning, ICLR 2018

[2] Jiayuan Mao, Chuang Gan, Pushmeet Kohli, Joshua B. Tenenbaum, Jiajun Wu, The Neuro-Symbolic Concept Learner: Interpreting Scenes, Words, and Sentences From Natural Supervision, ICLR 2019

**Summary Of The Paper:**

This paper proposes to measure the reasoning capabilities of VQA models by training an "adversary player" that can manipulate the image and fool the evaluated models. Given a sample, the "adversary player" moves objects such that the groundtruth answer remains valid and the scene is in-distribution. If the evaluated VQA model produces a different output, the "adversary player" receives a positive reward and is then optimized with a reinforcement learning algorithm.

**Summary Of The Review:**

Overall, this paper proposes a feasible way to evaluate the reasoning abilities of any black-box VQA model. Although the experimental results haven't revealed new problems, the proposed method can inspect current largely pretrained vision-language models to see if they perform reasoning on various tasks. Thus I endorse acceptance.

---

> ### Author Response · Authors · 2021-11-18
> **Response to comments of Reviewer 8VmJ**
>
> Thank you for your review and agreement that our approach to evaluate models is a good research direction. Among many other things, we claim that Visual Reasoning architectures with near-perfect (and “super-human”) performance on CLEVR can still suffer from reasoning and perceptual “blind spots”, even in simple scenarios. Those issues are not well captured by the standard metrics and static datasets, which focus rather on the statistically good performance but ignore harder cases. As deep learning models seem to provide strong performance in such a setting, we propose other and more subtle ways of evaluating models.
>
> We are also not aware of any work that exposes the reasoning issues of models by doing scene manipulations. Most existing works test perceptual capabilities and operate in the pixel domain [1,2,3]. Such perturbations require access to sensory inputs of the models, e.g., RGB images. The resulting images might also not exist in the real-world (as each pixel is changed independently, which is not the case during the image formation where pixels are changed according to the causes in the scene). Even though such attacks pose a security risk, it is less clear if such perturbations expose perceptual problems (likewise to “illusion” a human perception might also be susceptible to pixel-based attacks if we were able to compute gradients over the human visual system). Hence, are they a good material to study perception or reasoning?
>
> Closer to our work are semantic perturbations such as [4].  However, they deal mostly with perceptual “blind spots” by manipulating appearance, luminosity, and colors. Most experiments also assume a differentiable renderer which causes the same issue as the “pixel-based” perturbations. Some experiments are using object displacements with a more realistic non-differentiable renderer, but are less successful. Our work focuses on perceptual and reasoning “blind spots” and doesn't assume any access to the model or its sensors. We focus on the most basic and in-domain manipulations such as object displacement keeping the object attributes the same (e.g., color).  We are also different in techniques. They propose zeroth-order optimization for the non-differentiable renderer, we use RL and cast the problem as a two-player game.
>
> We also managed to expose weaknesses of strong object-centric and program execution models with our methodology. Below we show results from table 2 (Page 8) from our submission.
>
> | Model | CLEVR Accuracy | Avg. Accuracy Drop (State) | Max. Accuracy Drop (State) |
> |-------|----------------|-----------------------|-----------------|
> | FiLM     | 96.2           | 83.98 (-12.7%)        | 48.10 (-50.0%)  |
> | IEP       | 96.9           | 80.52 (-13.6%)        |  48.45 (-50.0%)  |
> | TbD      | 99.1           | 94.04 (-5.1%)          |   69.37 (-40.0%)  |
> | Mdetr    | 99.7           | 93.71 (-6.0%)         |  59.82 (-40.0%)  |
>
>
> [1] Intriguing properties of neural networks, Christian Szegedy, et al. 2014\
> [2] Explaining and harnessing adversarial examples, Goodfellow et al. ICLR 2015\
> [3] Towards evaluating the robustness of neural networks. Carlini and Wagner. CoRR, 2016\
> [4] Adversarial Attacks Beyond the Image Space, Zeng et al. CVPR 2019

---

> > ### Comment · Reviewer_8VmJ · 2021-11-22
> > **Response to authors and clarification of my initial summary**
> >
> > Lots of previous works [1][2][3] have argued the reasoning ability of CNN-LSTM based VQA models. Although these works didn't manipulate the scene, they have shown that the VQA models tend to catch superficial relations between the inputs and outputs rather than reasoning the question, then propose a new dataset split to evaluate the reasoning ability.
> >
> > Thus, as I claimed in the initial review, I am not surprised that the CNN-LSTM based RN or FiLM have a large performance drop after adversary scene manipulation. Also, the TbD, MDETR, as well as the MAC and NS-CL mentioned before, are aware of this problem. These works focus on improving the reasoning ability, and they don't drop a lot on the manipulated scene.
> >
> >
> > I agree that the proposed method is a new tool to evaluate the reasoning abilities of VQA models. But the experimental results haven't revealed new problems in this field.
> >
> >
> > [1] Yash Goyal, Tejas Khot, Douglas Summers-Stay, Dhruv Batra, and Devi Parikh. Making the V in VQA matter: Elevating the role of image understanding in Visual Question Answering. In CVPR, 2017.
> >
> > [2] Sainandan Ramakrishnan, Aishwarya Agrawal, and Stefan Lee. Overcoming language priors in visual question answering with adversarial regularization. In NIPS, 2018.
> >
> > [3] Justin Johnson, Bharath Hariharan, Laurens van der Maaten, Li Fei-Fei, C. Lawrence Zitnick, and Ross Girshick. CLEVR: A diagnostic dataset for compositional language and elementary visual reasoning. In CVPR, 2017.

---

> > > ### Author Response · Authors · 2021-11-22
> > > **Lack of robustness (ours and prior work)**
> > >
> > > We think the nature of our findings is
> > > quite different from [1,2,3].\
> > > Specifically,  [1,2] deal mostly with language biases on VQA.
> > > VQA is open-ended and ambiguous, and hence VQA-models might not have
> > > seen a sufficient amount of clean data to learn concepts well.
> > > Instead, those models use shortcuts to solve training objectives.
> > >
> > > To mitigate that, [3] propose CLEVR that  "contains
> > > minimal biases" and no ambiguities. CLEVER is close-ended and a
> > > sufficient amount of data should solve it (CLEVR-models can reach even
> > > 99%).
> > > MAC, TbD and Mdetr push the performance using standard accuracy on
> > > CLEVR, which we show is quite misleading. Also, it is interesting that in the case of MDetr, its advantages come from stronger input representations rather than from explicit components responsible for reasoning.
> > >
> > > Despite that, both TbD and Mdetr are
> > > still susceptible to scene manipulations (TbD Avg: -5.1% / Max: -40.0% , Mdetr Avg: -6.0% / Max: -40.0%  Table 2 Page 8).
> > >
> > > Hence, we suspect prior work was not fully aware of those issues, and
> > > our findings are still intriguing. It is not language bias, not
> > > noisy, imperfect annotations, not perception, and even not the
> > > difficulty of the dataset.
> > >
> > > [1] Yash Goyal, Tejas Khot, Douglas Summers-Stay, Dhruv Batra, and Devi Parikh. Making the V in VQA matter: Elevating the role of image understanding in Visual Question Answering. In CVPR, 2017.\
> > > [2] Sainandan Ramakrishnan, Aishwarya Agrawal, and Stefan Lee. Overcoming language priors in visual question answering with adversarial regularization. In NIPS, 2018.\
> > > [3] Justin Johnson, Bharath Hariharan, Laurens van der Maaten, Li Fei-Fei, C. Lawrence Zitnick, and Ross Girshick. CLEVR: A diagnostic dataset for compositional language and elementary visual reasoning. In CVPR, 2017.

---

### Official Review · Reviewer_u1a9 · 2021-11-02

**Correctness:** 3
**Technical Novelty And Significance:** 3
**Empirical Novelty And Significance:** 3
**Recommendation:** 6
**Confidence:** 4

**Main Review:**

To this reviewer, this is an interesting paper, that passes the bar of acceptance in ICLR. The experiments are thorough and well-explained. The writing is clear and detailed and the results are interesting. However, there are a few concerns that prevent me from giving a better score. Details Below

Strengths:

S1. Thorough Experiments: The choice of both data-driven (RN, FiLM), ground-truth program based (IEP, TBD) as well as a host of experiments clearly show the efficacy of the proposed model. There is very little to complain about ablations, details, statistical tests, and thoroughness in experiments.

S2. Clear Writing: The writing is clear and easy to understand. It also has enough details to easily allow reproduction by an expert.

S3. Interesting Results: The results are not always surprising. As expected, the program-based models are more robust to adversarial attacks compared to "from-scratch" methods. Regardless, some of the results for several strong non-program-based models still come as a surprise to me and raise interesting questions about the tested algorithms as well as learning as a whole. To me, this is the biggest strength of the paper.

Weaknesses:

W1. Experimental Validation has some weird issues: Maximal scores are, as I understand it, measure the loss of accuracy in the "worst" minigame compared to overall accuracy. I think that is an incorrect way to report it. In my opinion, it should be compared to the accuracy in *THAT* particular mini-game as opposed to comparing it with average accuracy for the model. Please clarify this choice.

W2. Some details are missing: How are the manipulations done and are steps taken to avoid unintended issues, e.g., occlusions, that would make a question unanswerable/ambiguous?




**Summary Of The Paper:**

The paper proposes a black-box adversarial attack on models trained on the CLEVR dataset. CLEVR dataset is widely considered solved, especially with program synthesis-based models which seem to achieve near-perfect accuracy on CLEVR with a fairly small amount of training examples. Moreover, due to their synthetic and limited nature, it can be reasonably assumed that they would be able to answer *any* variation of CLEVR scenes within the bounds of the CLEVR-universe (Barring intentionally OOD test sets such as CoGENT). However, this paper shows that it is possible to fool all classes of the CLEVR model using an adversarial player that simply re-configures the scene (which should have the same answer as before). While the drops inaccuracy is not drastic, it is still an interesting finding, especially for program synthesis-driven methods. The paper also ends with an experiment about whether a purely data-driven approach can ever learn to be fully "robust".

**Summary Of The Review:**

Overall, this is a solid paper with really interesting results. There are a few issues, noted in weaknesses, that prevent me from giving it a higher score. I will happily assign a higher score after rebuttal if those points are clarified.


UPDATE: I have read the author's responses and all other reviews carefully. Like others, I had some concerns about scene occlusion, etc. issues, but the authors have satisfactorily answered that. I also appreciate the authors' commitment to update the paper according to the reviews. That being said, I do not think the revisions have made for any *new* information for me to change my scores. I still think this is an interesting paper, albeit a bit limited in scope. I tend to keep my score as is ( ~ a weak accept)

---

> ### Author Response · Authors · 2021-11-18
> **Response to comments of Reviewer u1a9**
>
> Thank you for your comments.
> We agree with the reviewer that our results are surprising, especially how semantically equivalent scene manipulation can change the answers of even the strongest CLEVR models on the seemingly “solved” problem.
> We believe our findings will open new research directions on
> 1) how to evaluate perceptual or reasoning capabilities of deep learning models.
> 2) introducing multi-agent systems to evaluate or learn perception in computer vision and studies on reasoning (currently in synthetic environments).
>  3) “Semantic perturbations” of perceptual or reasoning capabilities of the models (as opposed to pixel-based perturbations).
>
> We will respond for each of the 2 weakness points separately.
> #### **Weakness 1 (W1):**
> We measured models’ accuracy on each minigame, before and after the scene manipulation.
> For the maximal drop, models exhibited accuracy similar to the overall average. Thus, our results are in accordance with the reviewer’s view. We chose the representation of our results wrt. the average accuracy so that we can easily present all the results in the same table against the same baseline. However, we see the confusion and we will update the table in the main paper. For now, the updated table can be seen in Appendix B2, page 37, and its fragment below.
>
>
> |     |Model Accuracy       | Accuracy after Adversarial Manipulations     | |
> |------------------------------||---------------------------------------------------||
> | Model | Mini-game            | State   | Pixels                |
> | SAN   | 80.0 / 74.0 / 72.3   | 28.84 / 47.65 / 68.42 | 43.26 / 66.32 / 70.87 |
> | FiLM  | 100.0 / 98.0 / 96.4  | 48.10 / 75.61 / 90.81 | 86.58 / 92.44 / 94.75 |
> | RN    | 100.0 / 94.0 / 93.1  | 46.60 / 63.28 / 90.03 | 74.56 / 87.42 / 91.05 |
> | IEP   | 100.0 / 98.0 / 97.1   | 48.45 / 74.41 / 93.99 | 87.21 / 93.50 / 95.73|
> | TbD   | 100.0 / 100.0 / 99.4  | 69.37 / 91.17 / 95.53 | 99.10 / 98.00 / 98.80|
> | MDetr | 100.0 / 100.0 / 99.5 | 59.82 / 86.43 / 94.51 | 99.70 / 97.80 / 99.10 |
> | ST    | 100.0 / 97.0 / 97.2  | 77.44 / 92.63 / 95.15 | 91.96 / 95.05 / 95.83 |
>
> #### **Weakness 2 (W2):**
>
> ##### ***How manipulations are done?***
> Our Adversarial player receives its input (image - question pair) that comes from the original CLEVR dataset and suggests manipulations in the form of an object displacement.
> Those manipulations are encoded into arguments for the Blender Graphics Engine, which renders the new images according to the Adversarial Player’s directives.
> Instead of pixel, patch, or in-place image transformations that other methods use, we opt for the same rendering engine that was employed during the creation of the original CLEVR dataset and use the same rules. The usage of reinforcement learning (A2C algorithm) enables us to bypass the need for differentiability during the image-generation process.
>
>
> ##### ***What steps were taken to avoid unintended issues?***
> With unconstrained scene manipulation, we may run into some unintended issues where the answer is not consistent with the question or a scene description anymore or leads to ambiguities or heavy occlusions that are against the original CLEVR rules.
> Section 4 (Consistency and In-distribution) describes how those issues are mitigated, where we use scene enforcers to create scenes that are not only valid but also consistent with the original CLEVR methodology.
>
> We added more extensive analysis to revised Appendix B2 (pages 37-38). Here, we summarize it.
>
> The scene-constraint enforcer is responsible for the creation of a scene/image that
> respects all constraints and statistical properties of the original CLEVR dataset. We use the same codebase as the original CLEVR to be exact regarding all those rules. E.g., we ensure that the minimum distance between the object centers, the margins along with cardinal directions, and the minimal number of pixels observable per object are kept the same as in the original work. In this way, objects are not placed very close to each other, are not occluded by each other, and are placed inside the observers’ point of view.
>
> The question-relevance enforcer checks if the newly generated scene produces the same answer as the old scene. By not enforcing this, every perturbation will be falsely “successful” as the question (which is kept the same before and after the manipulations) might have a different answer in the newly generated scene.
>
> Scenes that are rejected by any of the enforcers are not used. In this way, the manipulated scenes follow the same methodology as the original CLEVR.

---

> > ### Comment · Reviewer_u1a9 · 2021-11-19
> > **Thanks for clarifying**
> >
> > Thanks for clarifying. I am glad to see that both of my issues are well-addressed. I have updated my review accordingly.

---

> > > ### Author Response · Authors · 2021-11-20
> > > **Few extra comments on the scope of our work**
> > >
> > > We are happy to see that we addressed the reviewers' concerns. Regarding new information added during the revision, we believe that our revised version has a few more interesting insights such as better disentanglement of reasoning failures from perceptual failures. Such a study has not been done previously to the best of our knowledge. Moreover, we also have added a few paragraphs on “visual fairness” accompanied with the human study ( Appendix B5 - Pages 46-47).
> > >
> > > Regarding the scope of the work. We conducted experiments on CLEVR -- a diagnostic dataset for visual reasoning -- but our methodology is not limited to that. We chose CLEVR mainly due to 1) simplicity that would help in our analysis, 2) the dataset is fairly popular and hence there are many points of comparison. Both points help in the interpretation of the results. Otherwise, the methodology is versatile. It is model-agnostic and doesn’t require activations, gradients, or even access to pixels. From our viewpoint, it suffices if the tested model has produced correct or incorrect results. We also base our methodology on the Reinforcement Learning framework, which is nowadays fairly accessible. Hence, it could potentially also be used to test programs, or future robots only based on the behavioral outputs. We believe that the potential scope of our work is vast.

---

### Official Review · Reviewer_ELDV · 2021-11-06

**Correctness:** 3
**Technical Novelty And Significance:** 2
**Empirical Novelty And Significance:** 3
**Recommendation:** 6
**Confidence:** 4

**Details Of Ethics Concerns:**

Don't see ethical concerns for this work.

**Main Review:**


**Prior work**
* **Prior works about adversarial evaluation**: This type of adversarial analysis is not new in ML for QA models in particular (e.g. has been explored for reading comprehension over SQuAD and also over real-images) but it still interesting and unexplored in the context of CLEVR.
* **Prior work about consistency**: The particular sort of adversarial analysts that the paper explores is actually about measuring consistency - asking the same question over different scenes that hold the same relevant relations to make the question's answer unchanged, and checking wether the model indeed keeps its answer consistent over the different scenes. Such sort of evaluation has been proposed e.g. as part of the GQA dataset a couple of years ago, by pre-computing variety of semantically-equivalent questions and then measuring the consistency over them.
* **Related work**: The related work section gives a good coverage though in describing prior papers in related fields.

**The approach**
* **Problem statement**: The authors propose a clearly-defined approach and present it with a good degree of detail and preciseness.
* **adversarial scene manipulation**: A strength of the approach is that compared to other adversarial experiments that tend to change the inputs to be valid but not natural (e.g. a random or unreadable sequence of words that fails a textual QA model, or changes to pixel values that impact a visual models abut can't be noticed by a person), the sort of adversarial modifications performed by the proposed approach are completely natural and explainable (manipulating objects within the scene).
* **Perceptual vs. reasoning failures**: At the same time, I do worry about whether maybe some of the adversarial modifications are potentially just too visually-difficult for the model rather than really exposing real reasoning failures. There are multiple causes the could lead for that: (1) moving an object to be too occluded by another object so to be hard to identify, (2) or moving an object to be near another object of the same color in a particular position that will make the model harder to distinguish between them, (3) or making objects too similar in x or in y position such that it's hard to determine if one it behind/in front / to the left/right of another, or (4) putting a metallic object at a very particular position in the scene and/or partially occluded such that it will be hard to notice it's metallic reflection. Some of the qualitative examples indeed look like that is potentially the source for the change in the model answer.  I think there are 2 ways to mitigate this potential issue: First, constrain the adversarial model so to prevent it from making valid but "visually unfair" modifications (e.g. making object too close to each other etc), and second quantitatively measuring how the modification impact the performance of a pre-trained CLEVR object detector. On the standard CLEVR data a pre-trained detector can reach almost a perfect accuracy, and in the cases its performance will go down too due to the adversarial modification, we will know that the source of the failure was in perception rather than in reasoning. Overall I recommend the authors to explore these options further in order to give isolate the reasoning failures and strengthen the claims in the paper.
* **Other datasets**: Another important way to increase the confidence that the failures happen due to reasoning and not perception, which is always a good practice in general - is to test the approach also on another dataset. e.g. the NLVR or other 2d scenes could be good candidates, since their perceptual elements is easier than in clevr and so they are less prone to adversarial perception failures.
* **Adversarial vs. pre-computing questions**: I actually think that the pre-computed version as in GQA potentially allows simpler means for carrying out the consistency evaluation (by simply testing a trained model over a set of test questions). On the other hand, the adversarial conditions do enable looking for questions that will be especially hard for the particular evaluated model -- on a sense these settings therefore are more challenging, since they focus on looking for particular blind spots of the model, rather than measuring its consistency over randomly sampled equivalent scenes -- i.e. it could be that out of the space of semantically equivalent scenes, the model stays consistent in 95% of them, and the adversarial model finds those few rare cases it doesn't, not giving quantitative indication to how rare these cases are. Random sampling of multiple equivalent scenes would in contrast give a better estimation of the degree of consistency. Overall, I think while each approach has positives and negatives, it could be nice to include comparison in model's performance in the adversarial vs. randomly sampled equivalent scenes settings, in order to give a bigger more detailed picture about models' degree of consistency.
* **Black-box evaluation**: The black-box aspect of the approach -- the fact that it measures models performance and reasoning capabilities just by looking at their textual answers can be both a strength and a weakness: on the one hand, it make the evaluation more easily performed on a wider variety of models, but on the other, it gives a narrow view into why models fail when they do. Especially for CLEVR, there are multiple models that are e.g. based on  computing attention maps, executable programs, structured representations and other indications for their ongoing reasoning rather than just predicting an answer. I think it will be good to pick couple of such models and explore in the paper how they perform internally when failing vs. when being consistent -- are the any interesting patterns in the computed programs or attentions when models fail? Are there particular settings, scenes or questions that make models to be particularly vulnerable? having such sort of analysis would help a lot getting further insight not just into whether models are good or bad reasoners, but also for why are they not good enough and what could be done to improve them.

**Writing quality**
* The writing clarity of the paper is good, the idea and content are easy to follow, the motivation is clearly explained, and there are good visualizations that help understanding.
* **Spaces**: One small note regarding presentation of the manuscript: I guess due to space constraints there are no paragraphs at all -- the whole paper after the first couple pages is a one long block of text -- which makes it harder to read visually, so strongly recommend getting it back to the standard format in terms of spaces etc.


**Summary Of The Paper:**

The paper explores measuring VQA models under adversarial settings, by having two competing models, when reasoning over a CLEVR scene as usual, and the other seeks to make the scene more challenging to answer. These settings reveal the weaknesses of visual reasoning models, and demonstrate that their apparent almost perfect performance on the standard CLEVR dataset may give imprecise impression about their “true” reasoning capabilities, as the adversarial settings may better reveal.

Update: Following the detailed and thorough response by the authors that addressed most of my concerns, I'm happy to increase my score.

**Summary Of The Review:**

Overall, I think the paper proposes an interesting evaluation but is still missing empirically in multiple aspects that could strengthen the paper:
* Tracking potential reasons for the failure source (perceptual or reasoning) and making sure to reduce ways for the adversarial model to place objects in ways that make them too hard from the perceptual side to answer correctly.
* Comparing adversarial to randomly-sampled scenes to measure the degree of consistency given a question rather than penalize the model for the existence of a single way per question to fool it.
* Explore the model at least one more additional environment (e.g. 2d)
* Explore additional indications about models behavior when they do and do not fail, to give further insight about potential reasons for why they are in/consistent in different cases.

I therefore give the paper a marginal score, and encourage the authors to update the paper to make the evaluation more extensive!

---

> ### Author Response · Authors · 2021-11-20
> **Response to comments of Reviewer ELDV Part 1**
>
> We thank you for the review and critical analysis of our work. We agree that one among other contributions of our work is showing that standard, static evaluation of neural models is insufficient, and we need to re-think how to design our benchmarks. Benchmarks are among the driving forces to the development of new methods, and thus the progress of the whole field.
> Below, we address individual criticism.
>
> ***Prior work to adversarial evaluation:***
>
> Some recent works have observed issues with static evaluation and proposed more dynamic benchmarks, e.g., with a human-in-the-loop [1,2]. However, our work differs in that it is focused on visual reasoning, is interactive, and is fully automatic (where an RL agent replaces that “human” -in-the-loop).
>
>
>
> ***Similarities to GQA:***
>
> Authors behind GQA [3] have noticed that the VQA-models lack some reasoning robustness, e.g. models are not consistent wrt. the entailment. However, their approach to measuring reasoning is mostly linguistic. Moreover, GQA is still a static dataset, and we believe it will eventually fall under the same issues as other VQA-related datasets. Indeed, the current SOTA on GQA is about 76% -- quite close to SOTA on VQA. Therefore, we argue for automatic and interactive evaluation, where “adversarial” models could find and next expose various biases of the existing models. Hence, we redefine Visual Question Answering as a two-player game between two models, where one is trying to find biases or other weaknesses of the tested model. We also use drop and consistency scores. The latter is indeed similar to what GQA has proposed, but they measure consistency between different modifications of questions like entailment, whereas we measure the consistency when the scene is manipulated. Unlike GQA, we also use that score as a reward to our RL agent (Adversarial-Player).
>
> [1] Adversarial NLI: A New Benchmark for Natural Language Understanding, Nie et al. ACL 2020.\
> [2] Genie: A Leaderboard for Human-in-the-Loop Evaluation of Text Generation, Khashabi et al. CoRR 2021\
> [3] GQA: A New Dataset for Real-World Visual Reasoning and Compositional Question Answering Hudson et al. CVPR 2019.
>
> ***Disentangling reasoning from perceptual failures***
>
> As the reviewer suggested, in order to test quantitatively if the adversarial manipulations conditioned to the visual constraints make the image indeed visually ambiguous, we used a pre-trained CLEVR object detector to measure the difference in performance between original and manipulated images. For that, we use “Slot Attention” (Appendix B6 Pages 47-48) with pre-trained weights [1]. It is an unsupervised method for object discovery that achieves almost perfect performance on CLEVR. In both cases, before and after manipulations, “Slot Attention” achieves the same performance. Moreover, we also noticed that the MDetr[2] detections - (Appendix B4 Pages 43-44) are equally good after the scene manipulation as before. This gives the evidence that reported failures are rather due to wrong reasoning than failures in perception.
>
> [1] Object-Centric Learning with Slot Attention, Locatello et al. NeurIPS 2020\
> [2] Modulated Detection for End-to-End Multi-Modal Understanding, Kamath et al. 2021

---

> ### Author Response · Authors · 2021-11-20
> **Response to comments of Reviewer ELDV Part 2**
>
> ***Visual fairness***
> As said in the ***Disentangling reasoning from perceptual failures*** section, we made an effort to disentangle reasoning from perceptual failures. However, here, we can add a bit more on the visual fairness for further clarification.
>
> We make sure that all generated images do not suffer from any of the 4 possible causes of visual unfairness the reviewer noted. This is done with the help of two modules, mentioned briefly in Section 4 (scene constraint enforcer module, question relevance enforcer).
> As reviewer 2 also asked a similar question, we copy-paste our answer for your convenience below. Moreover, as there is the possibility that the enforcers are not entirely visually fair, we have also conducted a human study where we tried to understand if manipulated scenes have similar complexity to the original ones, e.g., objects are not heavily occluded.  We have found that complexities are about the same and can rule out that adversarial manipulations create visually unfair scenes. We added more details in Appendix B5(Pages 46- 47)
>
> Our Adversarial player receives its input (image - question pair) that comes from the original CLEVR dataset and suggests manipulations in the form of an object displacement. Those manipulations are encoded into arguments for the Blender Graphics Engine, which renders the new images according to the Adversarial Player’s directives. Instead of pixel, patch, or in-place image transformations that other methods use, we opt for the same rendering engine that was employed during the creation of the original CLEVR dataset and use the same rules. The usage of reinforcement learning (A2C algorithm) enables us to bypass the need for differentiability during the image-generation process.
>
> With unconstrained scene manipulation, we may run into some unintended issues where the answer is not consistent with the question or a scene description anymore or leads to ambiguities or heavy occlusions that are against the original CLEVR rules.
> Section 4 (Consistency and In-distribution) describes how those issues are mitigated, where we use scene enforcers to create scenes that are not only valid but also consistent with the original CLEVR methodology.
>
> We added more extensive analysis to revised Appendix B2 (pages 37-38). Here, we summarize it.
>
> The scene-constraint enforcer is responsible for the creation of a scene/image that
> respects all constraints and statistical properties of the original CLEVR dataset. We use the same codebase as the original CLEVR to be exact regarding all those rules. E.g., we ensure that the minimum distance between the object centers, the margins along with cardinal directions, and the minimal number of pixels observable per object are kept the same as in the original work. In this way, objects are not placed very close to each other, are not occluded by each other, and are placed inside the observers’ point of view.
>
> The question-relevance enforcer checks if the newly generated scene produces the same answer as the old scene. By not enforcing this, every perturbation will be falsely “successful” as the question (which is kept the same before and after the manipulations) might have a different answer in the newly generated scene.
>
> Scenes that are rejected by any of the enforcers are not used. In this way, the manipulated scenes follow the same methodology as the original CLEVR.

---

> ### Author Response · Authors · 2021-11-20
> **Response to comments of Reviewer ELDV Part 3**
>
> ***Comparison against Randomly Generated Scenes***
>
> In order to investigate this, we conducted an experiment comparing our Adversarial Player against a Random Scene Generator (RSG) that creates semantically equivalent scenes by sampling manipulations from a Uniform distribution, with respect to the given question/answer (and uses enforcers to make everything valid). We present the results here:
>
> | Model | CLEVR Accuracy  | Adv.Player Accuracy (Drop %) | RSG Accuracy (Drop %) |
> |-------|-----------------|------------------------------|-----------------------|
> | SAN   | 72.1            | 61.86 (-14.2%)               | 71.59 (-0.7%)         |
> | FiLM  | 96.2            | 83.98 (-12.7%)               | 95.81 (-0.4%)         |
> | RN    | 93.2            | 80.52 (-13.6%)               | 92.73 (-0.5%)         |
> | IEP   | 96.9            | 84.30 (-13.0%)               | 96.70 (-0.2%)         |
> | TbD   | 99.1            | 94.04 (-5.1%)                | 99.10 (-0.0%)         |
> | MDetr | 99.7            | 93.71 (-6.0%)                | 99.70 (-0.0%)         |
> | ST    | 96.8            | 89.34 (-7.7%)                | 96.50 (-0.3%)         |
>
> As it can be seen, RSG is not particularly successful. This further supports the argument that models under test only seem to be robust, but have vulnerabilities that can effectively be exploited by our adversarial player. Even though there is some intelligent mechanism required to find such manipulations, and could potentially be seen as rare cases under the CLEVR distribution, there are still many such corner cases. Another question that arises is, if the current models are so good (statistically) on various benchmarks, then perhaps it is time to move away from that evaluation paradigm, and consider instead harder cases.
>
> ***Explore additional indications about models behavior when they fail - Visualisation***
>
> For this inquiry, we have set up an experiment in which for each model we analyse visually its inner working mechanisms under a normal and manipulated scene scenario. Since the tested  models operate under different assumptions and architectural designs, we choose to analyze:
> - Attention maps of Stack-Attention Networks.
> - Saliency maps of the pre-ultimate and ultimate convolutional blocks of FiLM.
> - Programs that are synthesized alongside with their soft-attention maps for each module block in the TbD network.
> - Objects that were detected by the MDetr model.
>
> By inspecting the results, we find out that MDetr model correctly detects objects in each of the required reasoning steps, a result that is in accordance with the Slot Attention experiment mentioned in the ***Disentangling reasoning from perceptual failures*** section. It also strengthens the argument on the disentanglement of reasoning from perception (correct MDetr directions suggest reasoning “blind spots”).
>
> In the other models, we have noticed two patterns in their behavior.
> Concentrated focus onto a single object, or dissolution of the model’s focus into other non-important objects for the question. For instance, the tested model does not seem to understand the logical hop it has to take onto another object and instead is wrongfully stuck to the first candidate (according to saliency or attention maps).
>
> Please also note that this analysis (through detections or saliency maps) is complementary to the behavioral analysis, where the scene is manipulated and a question is posed. It also requires some access to the model, and thus it works under stronger assumptions.
>
> Further information and visuals have been added to Appendix B4, pages 40-45.
>
> ***Explore additional environments***
> One of the main requirements and core novelty of our work lies in the complete lack of assumptions regarding the model under the test and the use of a non-differentiable rendering method of achieving adversarial testing. To make that possible, access to the dataset generation code / graphic engine is vital. Unfortunately, most Visual QA datasets consist of image question pairs with no provided codebase for their generation. This is also true for the suggested NLVR dataset, which is still an open competition and although very interesting, we can’t currently use it. Upon disclosure of the dataset generation code by the creators, we would be happy to include it in our experiments. However, as the reviewer has proposed this dataset mainly to disentangle perception from reasoning, we hope that our answers above already do a good job on that.

---

> ### Comment · Reviewer_ELDV · 2021-11-30
> **Thank you for the response!**
>
> Dear authors,
> thank you very much for your detailed response. It addresses most of my concerns and I'm therefore happy to increase my score.
> *Regarding the random scenes experiments though I actually interpret the results in the opposite direction: if random perturbations that keep the scenes semantically equivalent don't impact the performance of most models almost at all, it means they are quite consistent in their reasoning, which serves as an indication for good robustness (even if not under adversarial conditions).
> - Reviewer

---

### Decision · Program_Chairs · 2022-01-20

**Decision:**

Accept (Poster)

**Comment:**

Introducing an adversarial agent that re-configures the rendered scenes of CLEVR to demonstrate that models that appear achieve super-human performance are actually easily fooled due to their lack of ability to reason, provides a nice insight into limitations with existing approaches and correspondingly how we evaluate on some benchmarks.  There is a persistent concern that the results are only on CLEVR, and that the adversarial examples are not really disproving reasoning but rather issues with vision.  However, overall reviewers were generally positive about the aims the work.